# Wavelet Diffusion Neural Operator

**Peiyan Hu**[2§*]  **Rui Wang**[3§*]  **Xiang Zheng**[4§]  **Tao Zhang**[1]  **Haodong Feng**[1]
**Ruiqi Feng**[1]  **Long Wei**[1]  **Yue Wang**[5]  **Zhi-Ming Ma**[2]  **Tailin Wu**[1†]
[1]Department of Artificial Intelligence, School of Engineering, Westlake University,
[2]Academy of Mathematics and Systems Science, Chinese Academy of Sciences,
[3]Fudan University, [4]South China University of Technology, [5]Microsoft AI4Science
{hupeiyan,wutailin}@westlake.edu.cn, ruiwang18@fudan.edu.cn

## Abstract

Simulating and controlling physical systems described by partial differential equations (PDEs) are crucial tasks across science and engineering. Recently, diffusion generative models have emerged as a competitive class of methods for these tasks due to their ability to capture long-term dependencies and model high-dimensional states. However, diffusion models typically struggle with handling system states with abrupt changes and generalizing to higher resolutions. In this work, we propose Wavelet Diffusion Neural Operator (WDNO), a novel PDE simulation and control framework that enhances the handling of these complexities. WDNO comprises two key innovations. Firstly, WDNO performs diffusion-based generative modeling in the wavelet domain for the entire trajectory to handle abrupt changes and long-term dependencies effectively. Secondly, to address the issue of poor generalization across different resolutions, which is one of the fundamental tasks in modeling physical systems, we introduce multi-resolution training. We validate WDNO on five physical systems, including 1D advection equation, three challenging physical systems with abrupt changes (1D Burgers' equation, 1D compressible Navier-Stokes equation and 2D incompressible fluid), and a real-world dataset ERA5, which demonstrates superior performance on both simulation and control tasks over state-of-the-art methods, with significant improvements in long-term and detail prediction accuracy. Remarkably, in the challenging context of the 2D high-dimensional and indirect control task aimed at reducing smoke leakage, WDNO reduces the leakage by 78% compared to the second-best baseline. The code can be found at `https://github.com/AI4Science-WestlakeU/wdno.git`.

## 1 Introduction

Many systems across science and engineering are described by partial differential equations (PDEs). Simulating and controlling these PDE systems are fundamental tasks with numerous applications, including weather forecasting (Lynch, 2008), controlled nuclear fusion (Carpanese, 2021), astronomical simulation (Courant et al., 1967), and aviation (Paranjape et al., 2013).

With developments of neural networks, deep learning-based methods have emerged to address this problem (Li et al., 2021; Lu et al., 2021; Tripura & Chakraborty, 2022; Hu et al., 2022). Among them, diffusion generative models (Ho et al., 2020b) achieve impressive results in both simulation (Cachay et al., 2023; Price et al., 2023; Rühling Cachay et al., 2023) and control (Ajay et al., 2022; Chi et al., 2023; Wei et al., 2024). On the one hand, simulation and control tasks are typically long-term, where small variations in the early stage can have a long-term impact on the full trajectory, making their accurate prediction and control difficult. Diffusion models alleviate the long-term challenge by the noise-learning mechanism and recovering the full trajectory from a Gaussian distribution as a whole. Therefore, they can better capture long-term dynamics and generate coherent plans for certain goals (Janner et al., 2022; Chi et al., 2023; Wei et al., 2024). On the other hand, PDE dynamics are typically high-dimensional and nonlinear, and the diffusion model demonstrates strong capabilities in modeling complex high-dimensional data (Ho et al., 2022; Harvey et al., 2022; Vahdat et al., 2022; Li et al., 2024). See Appendix D for more related works.

---

*Equal contribution. §Work done as an intern at Westlake University. †Corresponding author.

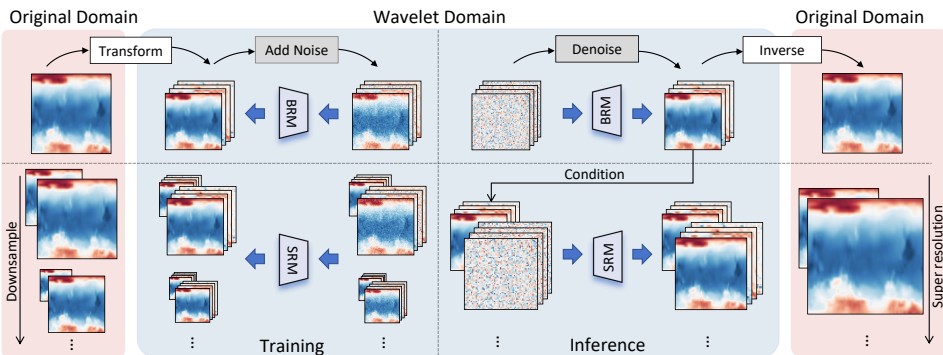

Figure 1: **Overview of WDNO.** The figure shows the training and inference of the Base-Resolution Model (BRM) and Super-Resolution Model (SRM).

However, for PDE simulation and control with diffusion models, two key challenges arise. Firstly, the evolution of physical systems is often accompanied by abrupt changes, which reflect key mechanisms of the system (Ben-Dor & Ben-Dor, 2007; Rassweiler et al., 2011). Due to their rapid and intense local variations, and even discontinuities, these changes are difficult to capture. Secondly, existing diffusion models typically operate on a fixed spatial-temporal resolution, and cannot generalize to finer resolutions (Croitoru et al., 2023; Yue et al., 2024; Shang et al., 2024), which is a fundamental requirement of neural PDE solvers (Li et al., 2021; Boussif et al., 2022; Yin et al., 2022).

In this work, we introduce Wavelet Diffusion Neural Operator (WDNO) to address the above two challenges. Our WDNO method consists of two key innovations: **(1) Generation in the wavelet domain.** Since the wavelet transform is both space and frequency localized and excels at approximating functions with *abrupt changes* (Tripura & Chakraborty, 2022), generation in the wavelet space endowed by the wavelet transform is ideal for modeling abrupt changes. Besides, due to the linearity and locality of the wavelet transform, it can integrate seamlessly with the multi-resolution training. **(2) Multi-resolution training.** To enable generalization to finer resolutions, we prepare training datasets across multiple spatial and temporal resolutions utilizing the approximate scale invariance. Since changes of the equation forms are approximately the same across resolutions, the model is trained to generalize to finer resolutions conditioned on coarser resolutions, which opens up the capability to generalize to even finer resolutions *not seen* during training.

Concretely, our contributions include the following: **(1)** We introduce the WDNO method that comprises diffusion in the wavelet space, addressing the challenges of modeling states with abrupt changes in simulation and control. **(2)** We propose multi-resolution training to address the issue of poor generalization to higher-resolution simulations, which is a fundamental task in PDE modeling. **(3)** We evaluate our method on 1D advection equation, complex PDEs with abrupt changes including 1D Burgers' equation, 1D compressible fluid, and 2D incompressible fluid, and a real-world dataset ERA5. Compared with strong baselines in physical simulation and control, our method shows competitive performance. Particularly, the 2D experiments are extremely challenging as they involve indirect control with 3,584 spatial control variables at each time step, for a total of 32 time steps. It is noteworthy that WDNO reduces 79% of the leaked smoke compared to the prior state-of-the-art.

## 2 PRELIMINARY

### 2.1 PROBLEM SETUP

We consider a PDE on $[0, T] \times D \subset \mathbb{R} \times \mathbb{R}^d$ with the following form

$$\frac{\partial u}{\partial t} = F\left(u, \frac{\partial u}{\partial x}, \frac{\partial^2 u}{\partial x^2}, \dots\right) + f(t, x), \quad (t, x) \in [0, T] \times D, \tag{1}$$

$$u(0, x) = u_0(x), \quad x \in D, \qquad B[u](t, x) = 0, \quad (t, x) \in [0, T] \times \partial D,$$

where $u : [0, T] \times D \to \mathbb{R}^n$ is the solution, with the initial condition $u_0(x)$ at time $t = 0$ and boundary condition $B[u](t, x) = 0$ on the boundary. $F$ is a function and $f(t, x)$ is the force term.

For such PDE systems, there are two fundamental tasks: simulation and control. The former involves learning a mapping from certain parameter functions $a$, such as initial conditions and boundary

conditions, to the solutions $u$ that represent a mapping from an infinite-dimensional function space to another infinite-dimensional function space. The latter task involves identifying the external control $f$ for a specific objective $\mathcal{J}(u, f)$ which is a function of $u$ and $f$, aiming at finding $f$ that minimizes $\mathcal{J}$.

## 2.2 DIFFUSION MODEL

A representative instance of diffusion models is the Denoising Diffusion Probabilistic Model (DDPM) (Ho et al., 2020b), which contains a forward and a reverse process to generate samples. In the forward process, noise is progressively added to clean data $\mathbf{x}_0$ until it is corrupted into Gaussian noise $\mathbf{x}_K \sim \mathcal{N}(\mathbf{0}, \mathbf{I})$. This process follows the Gaussian transition kernel $q(\mathbf{x}_{k+1}|\mathbf{x}_k) = \mathcal{N}(\mathbf{x}_{k+1}; \sqrt{\alpha_k}\mathbf{x}_k, (1 - \alpha_k)\mathbf{I})$, where $\{\alpha_k\}_{k=1}^K$ denotes the variance schedule. In the reverse process, data is sampled from Gaussian noise $\mathcal{N}(\mathbf{0}, \mathbf{I})$ and a denoising model $\boldsymbol{\epsilon}_\theta$ gradually removes the noise from the data until it returns the original clean data distribution. The model predicts the mean $\mu_\theta(\mathbf{x}_k)$ of $\mathbf{x}_{k-1}$ and the reverse process is defined with the transition $p_\theta(\mathbf{x}_{k-1}|\mathbf{x}_k) = \mathcal{N}(\mathbf{x}_{k-1}; \mu_\theta(\mathbf{x}_k, k), \sigma_k\mathbf{I})$.

To train the denoising model $\boldsymbol{\epsilon}_\theta$, the training loss is defined as follows, which optimizes a simplified variant of the variational lower-bound for the data's log-likelihood (Ho et al., 2020b).

$$\mathcal{L} = \mathbb{E}_{k \sim U(1,K), \mathbf{x}_0 \sim p(x), \boldsymbol{\epsilon} \sim \mathcal{N}(\mathbf{0},\mathbf{I})}[\|\boldsymbol{\epsilon} - \boldsymbol{\epsilon}_\theta(\sqrt{\bar{\alpha}_k}\mathbf{x}_0 + \sqrt{1 - \bar{\alpha}_k}\boldsymbol{\epsilon}, k)\|_2^2], \text{ where } \bar{\alpha}_k := \prod_{i=1}^k \alpha_i. \quad (2)$$

**Guided Diffusion Generation.** Modeling the conditional distribution $q(\mathbf{x}|\mathbf{y})$ enables controllable sample generation. Methods for conditioning in diffusion models include classifier-based guidance (Du et al., 2023) and classifier-free guidance (Ho & Salimans, 2022; Ajay et al., 2022). The former employs an additional classifier model trained on clean data to directly modify the denoising direction of the data during generation. The classifier-free conditioning simplifies the architecture and enables guided generation without an explicit classifier. It trains the model to learn both conditional and unconditional probabilities $\boldsymbol{\epsilon}_\theta(\mathbf{x}, \varnothing) \propto \nabla_{\mathbf{x}} \log q(\mathbf{x})$ and $\boldsymbol{\epsilon}_\theta(\mathbf{x}, \mathbf{y}) \propto \nabla_{\mathbf{x}} \log q(\mathbf{x}|\mathbf{y})$, where $\varnothing$ is an identifier that tells the model $\boldsymbol{\epsilon}_\theta$ to output $p(x)$ instead of $p(x|y)$ (Ho & Salimans, 2022). During sample generation, it combines noise terms following $\boldsymbol{\epsilon}_\theta(\mathbf{x}, \varnothing) + \omega(\boldsymbol{\epsilon}_\theta(\mathbf{x}, \mathbf{y}) - \boldsymbol{\epsilon}_\theta(\mathbf{x}, \varnothing))$, where $\omega \in [0, 1]$ is the weight. In this paper, we combine the use of both guidance methods.

## 3 METHOD

In this section, we detail our proposed WDNO from two perspectives: Section 3.1 describes how we perform the generative process within the wavelet domain, including basic concepts and practical implementation of wavelet transforms, and algorithms for applying WDNO to simulation and control problems. Section 3.2 presents the approximate scale invariance of PDE systems and our proposed multi-resolution training based on this property. The overall algorithm is presented in Figure 1.

### 3.1 GENERATION IN THE WAVELET DOMAIN

The WDNO performs generative control and simulation in the wavelet domain. Compared to the Fourier transform, the wavelet transform features locality while simultaneously retaining information in both space-time and frequency domains, allowing more accurate modeling for abrupt changes.

**Wavelet basis.** Intuitively, we use wavelet analysis to represent signals with basis functions localized in both space-time and frequency domains, taking values only within finite intervals. Specifically, this set of basis functions can be divided into two categories: one type is the scaling function $\phi$ used to represent the general outline (low-frequency information) of the original signal, and the other type is the mother wavelet $\psi$, which is used to depict the detailed information (high-frequency information) of the original signal (Alpert et al., 2002; Selesnick et al., 2005).

By scaling the function $\phi$ and mother wavelets $\psi$, we obtain $\phi_{l,m}$ and $\psi_{l,m}$:

$$\phi_{l,m}(x) = 2^{l/2}\phi(2^l x - m), \quad \psi_{l,m}(x) = 2^{l/2}\psi(2^l x - m),$$

where $m$ adjusts the position of the wavelet along the $x$-axis and $l$ represents the level of the basis. When $l$ increases, the wavelet narrows, and its frequency increases. Then, the entire space can be spanned by $\phi_{l,m}$ at a particular level $l_0$ and $\psi_{l,m}$ at levels greater than or equal to $l_0$, which can be presented as follows:

$$u(x) = \sum_m c_{l_0}(m)\phi_{l_0,m}(x) + \sum_{l=l_0}^{\infty}\sum_m d_l(m)\psi_{l,m}(x).$$

Thus we get the coarse wavelet coefficients $c_{l_0}(m)$ and the detail wavelet coefficients $d_l(m)$.

**Practical implementation.** However, due to the discrete nature of real-world data, the levels of $\psi_{l,m}$ will have an upper bound $L$, meaning there exists a minimum length interval for $\psi$. As mentioned in the introduction (Sec. 1), to preserve the locality of the data for integration with the multi-resolution training, we choose $l_0 = L$. So, the decomposition can be presented as:

$$u(x) = \sum_m c_L(m)\phi_{L,m}(x) + \sum_m d_L(m)\psi_{L,m}(x).$$

To verify the reliability of the wavelet decomposition's implementation, in Appendix A, we conduct tests of the reconstruction error on training data and discover that the relative $l_2$ errors of such reconstructions are on the order of $10^{-7}$, indicating that there is nearly no information loss. Further details about the wavelet transform can be found in Appendix A.

**WDNO for simulation.** For simulation, as introduced in Section 2.1, the objective is to learn a mapping from the equation parameter function $a$ to the solution function $u_{[0,T]}$. We can view the learning of this mapping as learning a conditional probability $p(u_{[0,T]}|a)$. However, we consider the conditional probability in the wavelet space, $p(W_{u_{[0,T]}}|W_a)$, where $W_u$ and $W_a$ are the wavelet-transformed values of $u_{[0,T]}$ and $a$. Here, we adopt classifier-free conditioning to guide the sampling process in diffusion models. Specifically, to ensure that the generated wavelet-transformed values $W_{u_{[0,T]}}$ align with the corresponding $W_a$, we include $W_a$ as a conditioning factor. Specifically, we initialize an optimization variable $W_{u_{[0,T]}}^{(k)}$ with Gaussian noise $\mathcal{N}(\mathbf{0},\mathbf{I})$, and iteratively update it via:

$$W_{u_{[0,T]}}^{(k-1)} = W_{u_{[0,T]}}^{(k)} - \eta\boldsymbol{\epsilon}_\theta(W_{u_{[0,T]}}^{(k)}, W_a, k) + \xi, \quad \xi \sim \mathcal{N}(\mathbf{0}, \sigma_k^2\mathbf{I}), \tag{3}$$

where $k$ denotes the denoising step, $\eta$ is the scaling factor and $\sigma_k$ represents the noise schedules. Repeatedly applying this denoising procedure from $k = M$ down to $k = 1$ yields the final solution $W_{u_{[0,T]}}^{(0)}$. Besides, during inference, we follow the Denoising Diffusion Implicit Model (DDIM) (Song et al., 2020), which can largely speed up the sampling process.

**WDNO for control.** For the control problem, in a task aimed to minimize $\mathcal{J}$, our goal is to find the optimal $f_{[0,T]}$ based on an environment determined by a parameter function $a$, such as the initial condition. Consequently, this problem can be naturally modeled as learning $p(f_{[0,T]}|a)$. Here, we also transform it into the wavelet domain, thus learning $p(W_{f_{[0,T]}}|W_a)$. Similar to the simulation, we employ a conditional diffusion model. However, a challenge arises in that we can only model and train $p(W_{f_{[0,T]}}|W_a)$ as represented in the training set, where $f$ is typically not optimal. To address this issue, we view the control problem from an energy optimization perspective, and thus during inference, we enhance the denoising process with guidance $\mathcal{J}$ to steer the generation of $f$ towards a smaller $\mathcal{J}$. Note that without this term, the model can only generate control sequences that follow the same distribution as the dataset, without optimizing for the control objectives. Specifically, initializing $W_{f_{[0,T]}}^{(k)}$ from Gaussian noise $\mathcal{N}(\mathbf{0},\mathbf{I})$, we iteratively update

$$W_{f_{[0,T]}}^{(k-1)} = W_{f_{[0,T]}}^{(k)} - \eta\left(\boldsymbol{\epsilon}_\theta(W_{f_{[0,T]}}^{(k)}, W_a, k) + \lambda\nabla_{W_{f_{[0,T]}}}\mathcal{J}(\hat{W}_{f_{[0,T]}}^{(k)})\right) + \xi, \quad \xi \sim \mathcal{N}(\mathbf{0}, \sigma_k^2\mathbf{I}), \tag{4}$$

where $\sigma_k$ and $\eta$ are the noise schedule and the scaling factor respectively, and $\lambda$ is the weight of guidance. Here $\hat{W}_{f_{[0,T]}}^{(k)}$ is the approximate noise-free $W_{f_{[0,T]}}^{(0)}$ estimated from $W_{f_{[0,T]}}^{(k)}$ by:

$$\hat{W}_{f_{[0,T]}}^{(k)} = (W_{f_{[0,T]}}^{(k)} - \sqrt{1-\bar{\alpha}_k}\boldsymbol{\epsilon}_\theta(W_{f_{[0,T]}}^{(k)}, W_a, k))/\sqrt{\bar{\alpha}_k}, \tag{5}$$

We calculate $\mathcal{J}$ in Eq. 4 based on $\hat{W}_{f_{[0,T]}}^{(k)}$ instead of directly using $W_{f_{[0,T]}}^{(k)}$ because otherwise noise in $W_{f_{[0,T]}}^{(k)}$ could bring errors to $\mathcal{J}$. Repeatedly applying this denoising procedure yields the final solution $W_{f_{[0,T]}}^{(0)}$. Similar to the simulation, we also employ the DDIM to accelerate the denoising.

## 3.2 MULTI-RESOLUTION FRAMEWORK

Next, to enable the diffusion model to generalize across different resolutions, we will introduce our multi-resolution framework based on the approximate scale invariance, which we will introduce in

the following. In contrast to the model mentioned in the previous section, which we refer to as the Base-Resolution Model (BRM), we will introduce a Super-Resolution Model (SRM) in this section. The framework integrates seamlessly with the wavelet transform technique and enables zero-shot super-resolution, which is one of the fundamental requirements of a neural operator.

**Approximate scale invariance.** We first introduce the approximate scale invariance. For simplicity, let us first assume that the spatial domain of the PDE in Eq. 1 is $D = [0, 1]$. Given the high-resolution data $d_+$ of size $N \times M$, and low-resolution data $d_-$ of size $(N/2) \times (M/2)$, although both are originally defined over the same spatiotemporal domain $[0, T] \times D$, we can rescale the low-resolution data into a new spatiotemporal domain $[0, T/2] \times \tilde{D}$, where the spatial domain $\tilde{D}$ is scaled to $[0, 1/2]$. In this case, $d_+$ and $d_-$ can be *aligned* to the same precision. However, note that the coordinates of $d_-$ are now scaled, meaning that the system no longer follows the original equation.

For any arbitrary spatial domain, we can always achieve such alignment through a linear transformation. We denote the linear transformations of time and space as $a_1 t + b_1$ and $a_2 x + b_2$ respectively. Then the stretched function actually satisfies the transformed version of the original equation:

$$\frac{\partial u}{a_1 \partial t} = F\left(u, \frac{\partial u}{a_2 \partial x}, \frac{\partial^2 u}{a_2^2 \partial x^2}, \dots\right) + f(t, x), \quad (t, x) \in [0, T/2] \times \tilde{D}. \tag{6}$$

Note that if we consistently consider the same factor of resolution change, this linear transformation remains constant, meaning that the coefficients $a_1$ and $a_2$ are fixed. Therefore, the pattern of change between different resolutions is consistent. Additionally, since the wavelet transform is linear and localized, this pattern remains consistent in the wavelet domain.

Correspondingly, in practical operations, we consider that each refinement of the discrete observations of the physical system follows the same pattern, which inspires us to develop the idea of multi-resolution training. Specifically, based on the training dataset at a given resolution, we downsample it to create a multi-resolution training dataset and then use this dataset for training to learn this pattern. Thus, during inference, we can naturally follow this pattern to achieve zero-shot super-resolution.

**Multi-resolution training data.** In practical implementation, we introduce the Super-Resolution Model, which is a conditional diffusion model. Assuming the resolution of the original training dataset is $N \times M$, that is, $N$ time steps and $M$ spatial points, we obtain data at the resolution of $(N/2) \times (M/2)$ through *downsampling*, which means we do not need finer-resolution data. We thus get the data pairs of sizes $N \times M$ and $(N/2) \times (M/2)$. This downsampling process can be repeated to obtain data pairs of $(N/2) \times (M/2)$ and $(N/4) \times (M/4)$, $(N/4) \times (M/4)$ and $(N/8) \times (M/8)$, and so forth, to compose the multi-resolution training dataset for training the Super-Resolution Model.

**Training.** We take the conditional diffusion model (Ho & Salimans, 2022) to model the conditional probability $p(W_h \mid W_l, W_{a_h})$, where $h$ and $l$ respectively present high- and low-resolution data of data pairs in the multi-resolution training dataset, $a_h$ is the high-resolution equation parameter, and $W_h$, $W_l$ and $W_{a_h}$ are the corresponding wavelet-transformed values. In detail, to align low-resolution with high-resolution data, we duplicate the low-resolution data to match the size of high-resolution data. During training, each batch randomly selects data pairs from a given resolution.

**Inference.** During the inference process, when super-resolution is required, we first downsample the high-resolution equation parameters $a$ to the same resolution $N \times M$ as the training data and perform a wavelet transform. Then, using the Base-Resolution Model, we first generate the wavelet coefficients of the base low resolution. Subsequently, we utilize the Super-Resolution Model to generate the data based on both the wavelet coefficients of lower-resolution results with size $N \times M$ and the wavelet coefficients of $a_h$ at the post-super-resolution resolution $2N \times 2M$. This process is iterated, allowing us to ultimately generate results with the same resolution as the original $a$.

## 4 EXPERIMENTS

In this section, we aim to test **(1)** the advantages of WDNO in handling complex long-term dynamics with abrupt changes on simulation and control problems, **(2)** the effectiveness of multi-resolution training in performing zero-shot super-resolution, and **(3)** the benefits of integrating wavelet transform.

We report the Mean Squared Error (MSE) measured on entire state sequences excluding initial conditions for the simulation tasks, and the control objective $\mathcal{J}$ for control problems. Besides, we consider state-of-the-art baselines from different fields. For control tasks, the following methods are

compared: **(1)** the classical control algorithm Proportional-Integral-Derivative (PID) (Li et al., 2006) **(2)** Supervised Learning method (SL) (Hwang et al., 2022), reinforcement learning and imitation learning methods including **(3)** Soft Actor-Critic (SAC) (Haarnoja et al., 2018), **(4)** Behavior Cloning (BC) (Pomerleau, 1988), **(5)** Behavior Proximal Policy Optimization (BPPO), and **(6)** DDPM (Zhuang et al., 2023). For simulation, we consider **(1)** DDPM (Ho et al., 2020a), **(2)** Wavelet Neural Operator (WNO) (Tripura & Chakraborty, 2022), **(3)** Multiwavelet Neural Operator (MWT) (Gupta et al., 2021), **(4)** Fourier Neural Operator (FNO) (Li et al., 2021), **(5)** CNN (Hwang et al., 2022), (6) Operator Transformer (OFormer) (Li et al., 2023), and **(7)** U-Net (Ronneberger et al., 2015). Details can be referenced in Appendix I, J and K. For reproducibility, the code is available here.

## 4.1 1D BURGERS' EQUATION

**Experiment setting.** We first consider the 1D Burgers' equation, a fundamental equation describing shock waves and turbulence in fluid dynamics, with the Dirichlet boundary condition and external force $f$, which follows previous works (Hwang et al., 2022; Mowlavi & Nabi, 2023) and is more difficult due to the long time horizon of 81 steps. The visualizations are presented in Figure 6. More details about the setting are in Appendix F. The simulation task is to learn the mapping from the initial condition $u_0$ and force term $f$ to the entire trajectory $u_{[0,T]}$, while the control objective $\mathcal{J}$ corresponding to the target state $u^*(x)$ and the fixed weight $\alpha$ is

$$\mathcal{J} = \int_D |u(T,x) - u^*(x)|^2 \mathrm{d}x + \alpha \int_{[0,T] \times D} |f(t,x)|^2 \mathrm{d}t \mathrm{d}x. \tag{7}$$

**Data preparation.** We perform a 2D wavelet transform on the original data using the *bior2.4* wavelet basis and the 'periodization' mode, implemented using the `pytorch_wavelets` package (Cotter, 2019). Since the initial condition and the target state are 1D, we take the 1D wavelet transform, repeat the coefficients, and then concatenate them with the 2D coefficients.

**Results** We report results of simulation and control tasks in Table 1 and Table 2a. From Table 1, it is evident that WDNO and DDPM achieve results that far surpass other baselines in simulation, demonstrating the capability of diffusion models for long-term predictions. In this particular simulation experiment, the performance of WDNO and DDPM is quite similar, while advantages of WDNO over DDPM are detailed in Section 4.6 and Section 4.7. For the control problem, WDNO achieves the best results, which clearly illustrates the superior performance of WDNO.

Table 1: **Results of simulation.** Bold font denotes the best model and the runner-up is underlined.

| Methods | 1D | | | 2D | |
|---|---|---|---|---|---|
| | Burgers' | Advection | Navier-Stokes | Fluid | ERA5 |
| WNO | 0.00572 | 4.216e-02 | 6.5428 | 0.07975 | – |
| MWT | 0.00052 | 3.468e-04 | 1.3830 | 0.01556 | 21.85750 |
| OFormer | 0.00023 | 1.858e-04 | 0.6227 | 0.04303 | 18.26230 |
| FNO | 0.00015 | 9.712e-04 | 0.2575 | 0.00684 | 14.38638 |
| CNN (1D) / U-Net (2D) | 0.00198 | 5.033e-04 | 12.4966 | 0.00737 | 15.51342 |
| DDPM | **0.00013** | 4.209e-05 | 5.5228 | 0.01578 | 15.21103 |
| **WDNO (ours)** | 0.00014 | **2.898e-05** | **0.2195** | **0.00231** | **12.83291** |

## 4.2 1D ADVECTION EQUATION

**Experiment setting.** Next, we consider the advection equation, which models pure advection behavior without nonlinearity. This dataset, sourced from `PDEBench` (Takamoto et al., 2022), is set up to predict 80 timesteps of evolution based on the one-time-step initial condition. The system exhibits relatively smooth and simple dynamics. We aim to observe the performance of various methods on a system without abrupt changes using this dataset.

**Data preparation.** Since the data shape is similar to that of the 1D Burgers' equation, the data preparation process is consistent with that of the first experiment.

**Results.** From results in Table 1, we can observe that most models achieve low prediction errors. However, WDNO still delivers the best results.

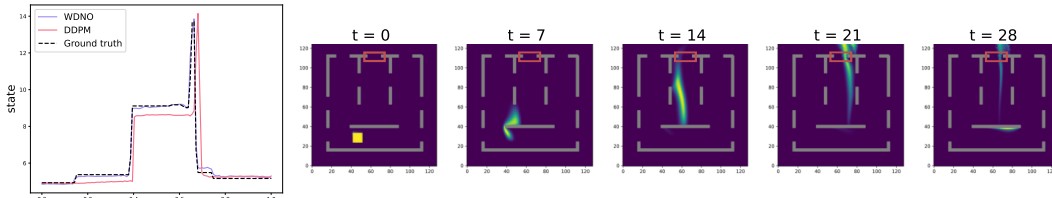

(a) **Results of WDNO and DDPM on the 1D Navier-Stokes equation.**

(b) **Results of WDNO on the 2D indirect control.** The objective is to navigate the yellow smoke to get around grey obstacles and reach the target bucket located at the top center.

Figure 2: **Visualizations of 1D Navier-Stokes equation and 2D incompressible fluid.**

### 4.3 1D Compressible Navier-Stokes Equation

**Experiment setting.** We also consider the important 1D compressible Navier-Stokes equation which can describe complex phenomena, such as shock wave formation and propagation in aerodynamics around airplane wings and interstellar gas dynamics. We consider a particularly challenging scenario from the 1D CFD dataset in `PDEBench` (Takamoto et al., 2022). We select extremely small viscosity coefficients, $\eta = 10^{-8}$ and $\zeta = 10^{-8}$. The initial conditions are shock-tube fields consisting of piecewise constant values generating shocks and rarefactions. Boundary conditions allow waves to exit the domain. Since this pre-existing dataset does not include time-varying control terms, we only perform the simulation task on it. We provide more details in Appendix G.

**Data preparation.** The data preparation process is also similar to the above ones.

**Results.** From Table 1, we can observe that WDNO still gains the best performance among strong baselines. It is particularly noteworthy that the MSE of DDPM exceeds that of WDNO by over 25 times. In Figure 2a and Figure 7, we further present the detailed prediction results of DDPM and WDNO. It can be seen that for physical dynamics with abrupt changes, DDPM struggles to model shocks and loses many fine details. This highlights the necessity of introducing the wavelet transform. More results, including MSEs, MAEs, and $L_\infty$, and other baselines can be found in Appendix C.1.

Table 2: **Results of control tasks.** Bold font denotes the best model and the runner-up is underlined.

<table>
<tr><td colspan="2" align="center">(a) 1D Burgers' equation.</td><td colspan="2" align="center">(b) 2D incompressible fluid.</td></tr>
<tr><td>Methods</td><td>$\mathcal{J}$</td><td>Methods</td><td>$\mathcal{J}$</td></tr>
<tr><td>PID (surrogate-solver)</td><td>0.6645</td><td>BC</td><td>0.3085</td></tr>
<tr><td>SAC (pseudo-online)</td><td>0.1376</td><td>BPPO</td><td>0.3066</td></tr>
<tr><td>SAC (offline)</td><td>0.3210</td><td>SAC (pseudo-online)</td><td>0.3212</td></tr>
<tr><td>BC (surrogate-solver)</td><td>0.2998</td><td>SAC (offline)</td><td>0.6503</td></tr>
<tr><td>BC (solver)</td><td>0.1879</td><td>DDPM</td><td>0.3124</td></tr>
<tr><td>BPPO (surrogate-solver)</td><td>0.3075</td><td>WDNO (ours)</td><td>0.0679</td></tr>
<tr><td>BPPO (solver)</td><td>0.1867</td><td></td><td></td></tr>
<tr><td>SL</td><td>0.0235</td><td></td><td></td></tr>
<tr><td>DDPM</td><td>0.0272</td><td></td><td></td></tr>
<tr><td>WDNO (ours)</td><td>0.0205</td><td></td><td></td></tr>
</table>

### 4.4 2D Incompressible Fluid

**Experiment setting.** Next, we experiment on 2D fluid problems following the incompressible Navier-Stokes equation. The experiment setting, a complex scenario close to real-world, follows previous works (Wei et al., 2024), where the control can only be exercised out of the frame as shown in Figure 2b. The boundary condition at obstacles is the no-slip condition, meaning that the velocities are set to 0 at the boundary. This experiment thus includes fluid-solid coupling, where functions have discontinuities and are hard to model. The different data trajectories share the same initial velocity field; the variations are in initial smoke positions, specifically the smoke's initial density, and control

sequences. The simulation task is to predict the smoke's density, velocity field, and the percentage of smoke passing through the target bucket based on the initial smoke density and control sequences.

For the control problem, our goal is to move the smoke from its initial position, located beneath the central obstacle, into the middle bucket at the top. To be more specific, $\mathcal{J}$ is defined as the percentage of smoke not passing through the target bucket. Firstly, this objective presents considerable challenges due to the restriction that forces can only be applied in the peripheral regions. This problem requires the model to plan ahead in the middle of the entire trajectory to avoid entry into the wrong opening. Furthermore, we need to generate 3,584 control parameters over a time span of 32 steps in these peripheral zones to indirectly control the velocity field in the central region.

**Data preparation.** We perform a 3D wavelet transform on original data using *bior1.3* wavelet basis and 'zero' mode, implemented through `Pytorch Wavelet Toolbox (ptwt)` (Wolter et al., 2024). Since the initial condition and percentage of smoke are 2D and 1D respectively, we take the 2D and 1D wavelet transform and repeat the coefficients to concatenate them.

**Results.** Table 1 are the simulation results, showing our method is far superior to DDPM and exceeds all the baselines. It is worth noting that the prediction error of WDNO is an order of magnitude lower than that of DDPM. As for the results of the control problem shown in Table 2b, our method can make more than 90% of the smoke pass through the target bucket, and its $\mathcal{J}$ is 22% of the next best method's $\mathcal{J}$, showing our model's superiority under complex dynamics with abrupt changes.

### 4.5 ERA5

**Experiment setting**. The ERA5 dataset (Kalnay et al., 2018), provided by ECMWF, is a challenging real-world dataset for weather forecasting. It offers hourly atmospheric estimates with a 0.25° latitude-longitude resolution from the Earth's surface to 100 km altitude, spanning from 1979 to the present. We conduct simulation experiments on this dataset to demonstrate the superior performance of WDNO. The selected variable is temperature, and the specific task involves predicting the system's evolution over the next 20 hours based on its state over the past 12 hours.

**Data preparation**. Due to similar data size, the process of data preparation is similar to Section 4.4.

**Results**. We present the results in Table 1. Here we experiment with different parameters for WNO, but all configurations fail to converge. It is clear that WDNO still achieves the best performance, with a relative $L_2$ error as low as 0.0161, demonstrating its outstanding capability on challenging datasets.

### 4.6 ZERO-SHOT SUPER RESOLUTION

In this subsection, we will present the super-resolution simulation results for the 1D Burgers' equation and 2D incompressible fluid. For the 1D experiments, the resolution of the training dataset is of the time-space resolution $80 \times 120$. We demonstrate the results of single, double, and triple super-resolution steps on both time and space, with the corresponding unseen resolutions of $160 \times 240$, $320 \times 480$, and $640 \times 960$ respectively. For the 2D experiments, the training dataset has a resolution of $32 \times 64 \times 64$, and we transfer to the resolution $32 \times 128 \times 128$. The visualization of 1D zero-shot super resolution is presented in Figure 3.

To evaluate the performance across different resolutions, we interpolate the outcomes of each super-resolution step to the highest resolution level. This allows us to assess whether the model can accurately



Figure 3: **1D zero-shot super-resolution.** The first row shows WDNO's simulation results with no super resolution, one-level super resolution, and two-level super resolution. The second row is the ground truth, and the third row is the difference between the first and second rows. As resolution increases, WDNO's output gets closer to the ground truth, demonstrating its zero-shot super resolution capability.

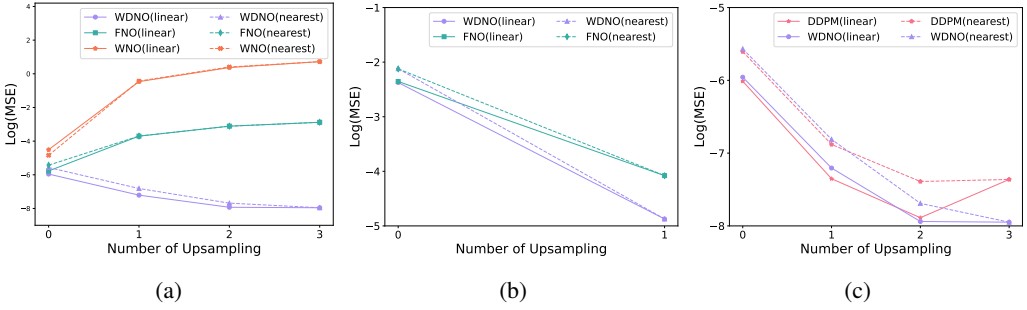

Figure 4: **Results in Section 4.6 and Section 4.7 after $n$ super-resolution steps.** All MSEs are calculated at the finest resolution through linear or nearest interpolation. (a), (b) are 1D Burgers' and 2D results in Section 4.6, respectively, and (c) is the results in Section 4.7.

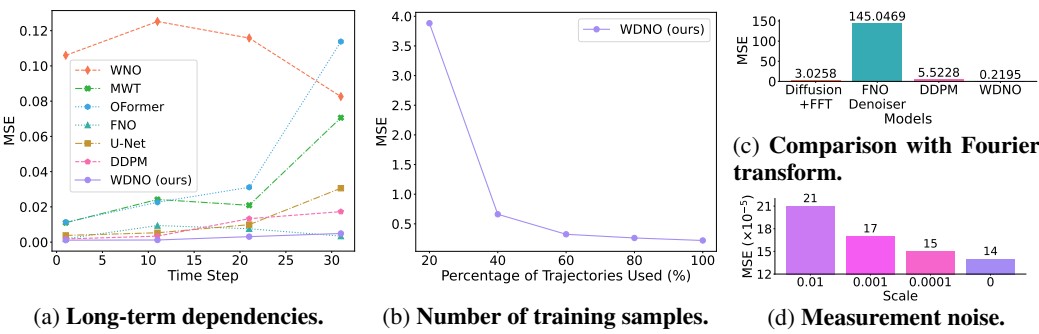

Figure 5: **Results of ablation studies.**

generate data on finer grid points beyond the resolutions encountered during training. We consider linear interpolation and nearest interpolation, taking the mesh-invariant model FNO and WNO as the baselines. Due to WNO's implementation, it can only perform spatiotemporal super-resolution simultaneously, making it unsuitable for 2D super-resolution experiments. As shown in Figure 4a, Figure 4b, Table 16 and Table 17, in both 1D and 2D scenarios, our method surpasses results of interpolation by achieving significantly improved outcomes with each super-resolution step. It can effectively reconstruct the values on the newly added grid points at the highest resolution, outperforming the mesh-invariant FNO and WNO.

## 4.7 ABLATION STUDY

**Abrupt changes.** We first verify whether the wavelet transform can enhance DDPM's ability to model abrupt changes. To this end, we present the system's states and prediction errors of WDNO and DDPM over time in Figure 6 and Figure 9. Note that, although WDNO and DDPM have similar overall MSEs in Table 1, we can observe that at moments when the state exhibits abrupt changes in space, WDNO achieves a lower prediction error compared to DDPM. This demonstrates that the wavelet transform helps to model dynamics with abrupt changes that are otherwise difficult to learn.

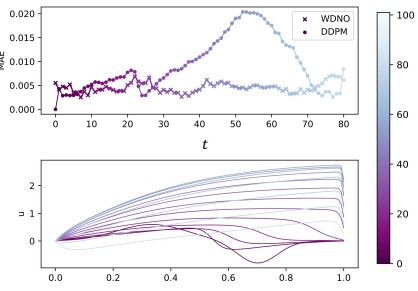

Figure 6: MAE and state trajectories.

**Combination of wavelet and multi-resolution training.**
To assess the efficacy of integrating wavelet transform with multi-resolution training due to the wavelet transform's locality, we provide outcomes from DDPM combined with multi-resolution training by applying the framework directly in the space-time domain, as depicted in Figure 4c. Notably, in the 1D experiment, as the number of super-resolution steps increases, evaluations at the highest resolution reveal that the disparity between WDNO and the application of the multi-resolution training in the original space-time domain becomes more pronounced, verifying the efficiency of utilizing wavelet transforms for super resolution.

**Comparison with Fourier transform.** We also evaluate the diffusion model in the Fourier domain (Diffusion + FFT). The implementation strictly follows WDNO, except for replacing the wavelet transform with Fourier transform. The MSEs on the 1D compressible Navier-Stokes equation are shown in figure 5c. While the Fourier transform also provides some improvement over DDPM, its performance is significantly inferior to that of the wavelet transform, which verifies that wavelet transforms inherently decompose information into low-frequency components and high-frequency details across different directions, making them more effective for learning complex system dynamics, such as those with abrupt changes. In addition, we take the FNO as the noise prediction model (FNO Denoiser), but the results indicate inferior performance. This may be because FNO tends to filter out high-frequency information, which is crucial for a noise prediction mode.

**Long-term dependencies.** Long-time predictions tend to perform poorly due to error accumulation and prediction instability. Therefore, capturing long-term dependencies allows WDNO to grasp the dynamics over extended periods better, naturally improving WDNO performance. To further verify it, in Figure 5a, we provide errors of baselines and WDNO at different time steps in the 2D simulation experiment. It is obvious that WDNO exhibits the slowest error growth, confirming its ability to capture long-term dependencies.

**Measurement noise.** To evaluate on datasets with increasing measurement noise, we add noise to both the training and testing datasets of 1D Burgers' equation, sampled as Gaussian noise scaled by the original data's standard deviation multiplied by a scale factor. We test scale factors of 0.01, 0.001, and 0.0001. As shown in the Figure 5d, WDNO 's results exhibit minimal variation with changes in scale, demonstrating its robustness to noise.

**Number of training samples.** We reduce the training dataset size to 0.2, 0.4, 0.6, and 0.8 times the current size (9000 samples) and measure WDNO's MSE on the 1D compressible Navier-Stokes equation. The results in Figure 5b show that even when the dataset size is reduced to 0.4 times, WDNO 's error remains within a relatively small range. When the dataset size is reduced to 0.2 times, the error shows a noticeable increase.

**Additional results.** Due to space constraints, we provide additional details in Appendix C, which include sensitivity analysis of key hyperparameters, verifying approximate scale invariance, evaluating the sensitivity of baselines and WDNO to noise in control sequences, comparing computational resource usage between baselines and WDNO, and analyzing the impact of the guidance parameter.

## 5 LIMITATION AND FUTURE WORK

Firstly, although we do not conduct real-world experiments, WDNO is not limited to the specific environments, which means that it can be applied to real scenarios, such as turbulence, structural materials and plasma, which we will leave as future work. Secondly, due to the wavelet transform and denoising model U-Net, WDNO is only applicable to static, uniform grid data. We are considering applying WDNO to irregular data by using geometric wavelets (Xu et al., 2018) combined with diffusion models designed for graph structures (Vignac et al., 2023), or projecting data from irregular grids onto regular uniform grids (Li et al., 2020b; Lin et al., 2023), among others. Finally, our current approach does not yet incorporate information from equations, such as adding physics-informed loss based on the PDEs, which can enhance the model's accuracy, robustness, and generalizability.

## 6 CONCLUSION

In this paper, we have introduced Wavelet Diffusion Neural Operator (WDNO), a method for simulation and control of PDE systems. By introducing two innovations of generation in the wavelet domain and multi-resolution training, WDNO addresses the challenges of modeling states with abrupt changes and generalizing across resolutions typical in PDE systems. Experiments on challenging settings including the 1D Burgers' equation, 1D Adevection Equation, 1D CFD, 2D incompressible fluid and ERA5 demonstrate WDNO's superior performance and its ability to generalize to much finer spatial and temporal resolutions than in training. We believe that WDNO will be useful for complex physical simulation and control in a wide range of scientific and engineering domains.

## 7 ACKNOWLEDGMENT

We thank Tengfei Xu and Tao Zhang for discussions and for providing feedback on our manuscript. We also gratefully acknowledge the support of Westlake University Research Center for Industries of the Future; Westlake University Center for High-performance Computing. The content is solely the responsibility of the authors and does not necessarily represent the official views of the funding entities.

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

## A  DETAILS OF WAVELET DECOMPOSITION

In this section, we provide a detailed introduction to wavelet transforms. Let $V_l$ be the space spanned by scaling functions $\phi_{l,m}$, $m \in \mathbb{Z}$, and $W_l$ be the space spanned by wavelets $\psi_{l,m}$, $m \in \mathbb{Z}$. The scaling functions possess two fundamental properties:

1. The scale function is orthogonal for its integer translation.

2. The wavelet spaces satisfy a nested and increasing sequence of spaces:

$$V_{-\infty} \subset \cdots \subset V_{-1} \subset V_0 \subset V_1 \subset \cdots \subset V_\infty.$$

As for the space $V_l$ and $W_l$, they have the following relationship:

$$V_{l+1} = V_l \oplus W_l.$$

Intuitively, the spaces $W_l$ spanned by the wavelet functions complement the missing information between the scaling function spaces of different levels.

Then, the process of wavelet transform can be viewed as convolving the scaling and wavelet functions of a certain level with the original signal, effectively splitting the signal into low-frequency and high-frequency components. Subsequently, the low-frequency part is further decomposed. This results in obtaining coefficients $c_{l_0}$ and $d_{l_0}, d_{l_0+1}, d_{l_0+2}, \ldots$.

There are numerous types of wavelet bases that have different waveforms. Here we provide further insights into the criteria used for wavelet selection. For the wavelets we consider (bior, db, sym), bior and sym wavelets offer symmetry, which reduces phase distortion during processing and allows for more accurate reconstruction compared to db, as also reflected in the reconstruction loss table. Regarding the choice of wavelet scale, despite the higher smoothness of higher-order wavelets, they generally have larger value ranges. Therefore, for data with small spatiotemporal size, high-order wavelets may not be suitable. For example, in the 1D data with a size of $N \times 81 \times 120$, we choose bior2.4, while for the 2D data with a size of $N \times 32 \times 64 \times 64$, we select bior1.3, a lower-order wavelet. Using wavelets with excessively large support lengths may distort coefficients near the boundaries and fail to effectively decompose signal details, hindering the effectiveness of multi-resolution decomposition.

Specifically, we select *bior2.4* and *bior1.3* from the Biorthogonal wavelet family for our experiments on the 1D Burgers' equation and 2D incompressible fluid, respectively. And we we use the 'periodization' mode in 1D and the 'zero' mode in 2D. Due to the presence of the temporal dimension, we perform a two-dimensional wavelet transform on data from the 1D Burgers' equation and a three-dimensional wavelet transform on data from the 2D incompressible fluid.

In Table 3, we report the reconstruction errors of wavelet transforms using different wavelet bases on the 1D Burgers' equation and 2D incompressible fluid, the results show that the reconstruction is significantly low.

Table 3: **Reconstruction relative $L_2$ errors on 1D Burgers' equation and 2D incompressible fluid.**

| Types of wavelet | 1D | 2D |
|---|---|---|
| bior1.3 | 1.09e-07 | 3.32e-07 |
| bior2.4 | 8.32e-08 | 2.65e-07 |
| db4 | 1.39e-07 | 4.39e-07 |
| sym4 | 1.17e-07 | 3.74e-07 |

Besides, in Table 4, we provide the total time consumption for Fourier and wavelet transforms on the training set of the 1D compressible Navier-Stokes equation. The Fourier transform is implemented using PyTorch's 2D Fast Fourier Transform function. Both times are recorded on an A100 GPU with a batch size of 2000. From the results, we can observe wavelet transform's efficiency.

Table 4: **Total time consumption for Fourier and wavelet transforms on the training set of the 1D compressible Navier-Stokes equation.**

|  | Wavelet transform | Fourier transform |
| --- | --- | --- |
| Time (s) | 1.0171 | 1.3810 |

# B    VISUALIZATION OF EXPERIMENT RESULTS

## B.1    VISUALIZATIONS OF 1D COMPRESSIBLE NAVIER-STOKES EQUATION

In Figure 7, we present visualizations of predictions from WDNO and DDPM. It is clear that WDNO is far better at modeling states with abrupt changes. While DDPM can not capture details, WDNO can successfully predict these precise changes.

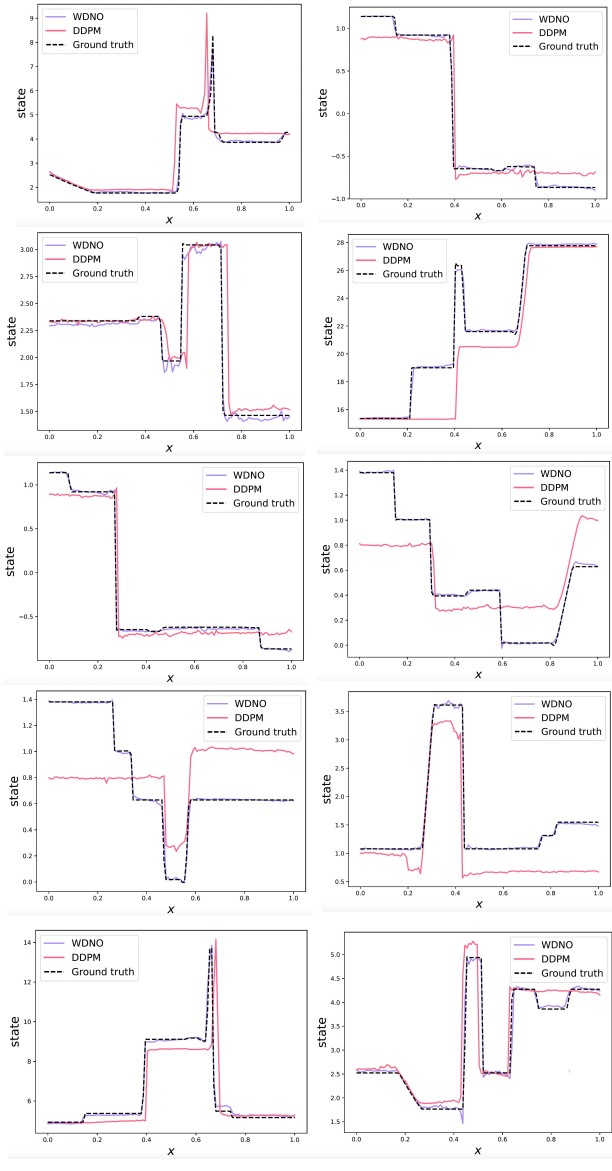

Figure 7: **Visualizations of WDNO's and DDPM's performance on simulation of the 1D compressible Navier-Stokes equation.**

### B.2 VISUALIZATIONS OF 2D INCOMPRESSIBLE FLUID

We provide visual results of WDNO on challenging 2D control tasks in Figure 8. It can easily be observed that, for many trajectories, our method successfully guides the smoke to pass essentially through the target bucket, which is not achieved by other baselines.

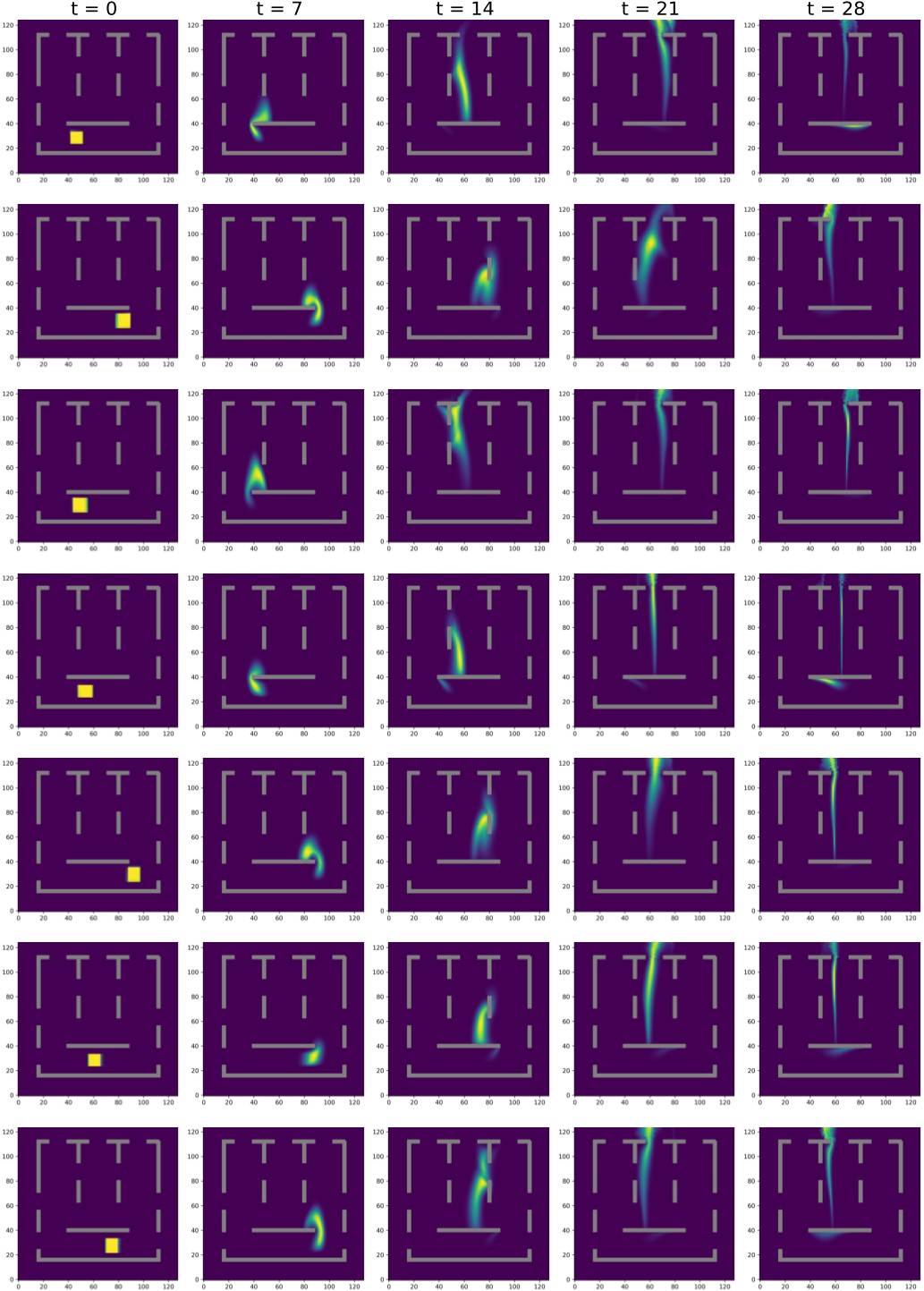

Figure 8: **Visualizations of WDNO's performance on the 2D incompressible fluid control task.**

## C ADDITIONAL RESULTS OF EXPERIMENTS

### C.1 MORE COMPARISONS ON 1D COMPRESSIBLE NAVIER-STOKES EQUATION

Here, we provide more results on simulation of 1D Compressible Navier-Stokes Equation. We further compare WDNO with Transolver (Wu et al., 2024a), CNO (Raonic et al., 2024), MSVI (Iakovlev et al., 2022), ACDM (Kohl et al., 2024), and DiffusionPDE (Huang et al., 2024). We also add comparisons with diffusion models in Fourier domain and FNO denoiser. The results in Table 5 demonstrate that WDNO still achieves the best performance on MSE. It can be observed that the trends of MAE align closely with MSE. However, the $L_\infty$ error values across different methods are relatively similar because this metric only considers the maximum value across the entire spatiotemporal domain, thus capturing less information.

Table 5: **Comparison of Various Models Based on Error Metrics**

| Model | MSE | MAE | $L_\infty$ **Error** |
|---|---|---|---|
| Transolver | 4.9984 | 0.4025 | **4.87284** |
| CNO | 0.3987 | 0.2765 | 9.9169 |
| MSVI | 1.7063 | 0.6047 | 17.0386 |
| ACDM | 4.6574 | 0.8946 | 60.9370 |
| DiffusionPDE | 5.5936 | 0.9792 | 16.0514 |
| WNO | 6.5428 | 1.1921 | 21.3860 |
| MWT | 1.3830 | 0.5196 | 11.3677 |
| OFormer | 0.6227 | 0.4006 | 30.9019 |
| FNO | 0.2575 | 0.1985 | 11.1495 |
| CNN | 12.4966 | 1.2111 | 17.6116 |
| DDPM | 5.5228 | 0.9795 | 16.0532 |
| Diffusion + FFT | 3.0258 | 0.8498 | 14.6670 |
| FNO Denoiser | 145.0469 | 6.6406 | 31.7515 |
| WDNO (ours) | **0.2195** | **0.1049** | 13.0626 |

### C.2 ABRUPT CHANGES

Here we provide more visualizations of the comparison between WDNO's and DDPM's MAE of different time steps. Figure 9 verifies that WDNO can better model abrupt changes due to the wavelet transform.

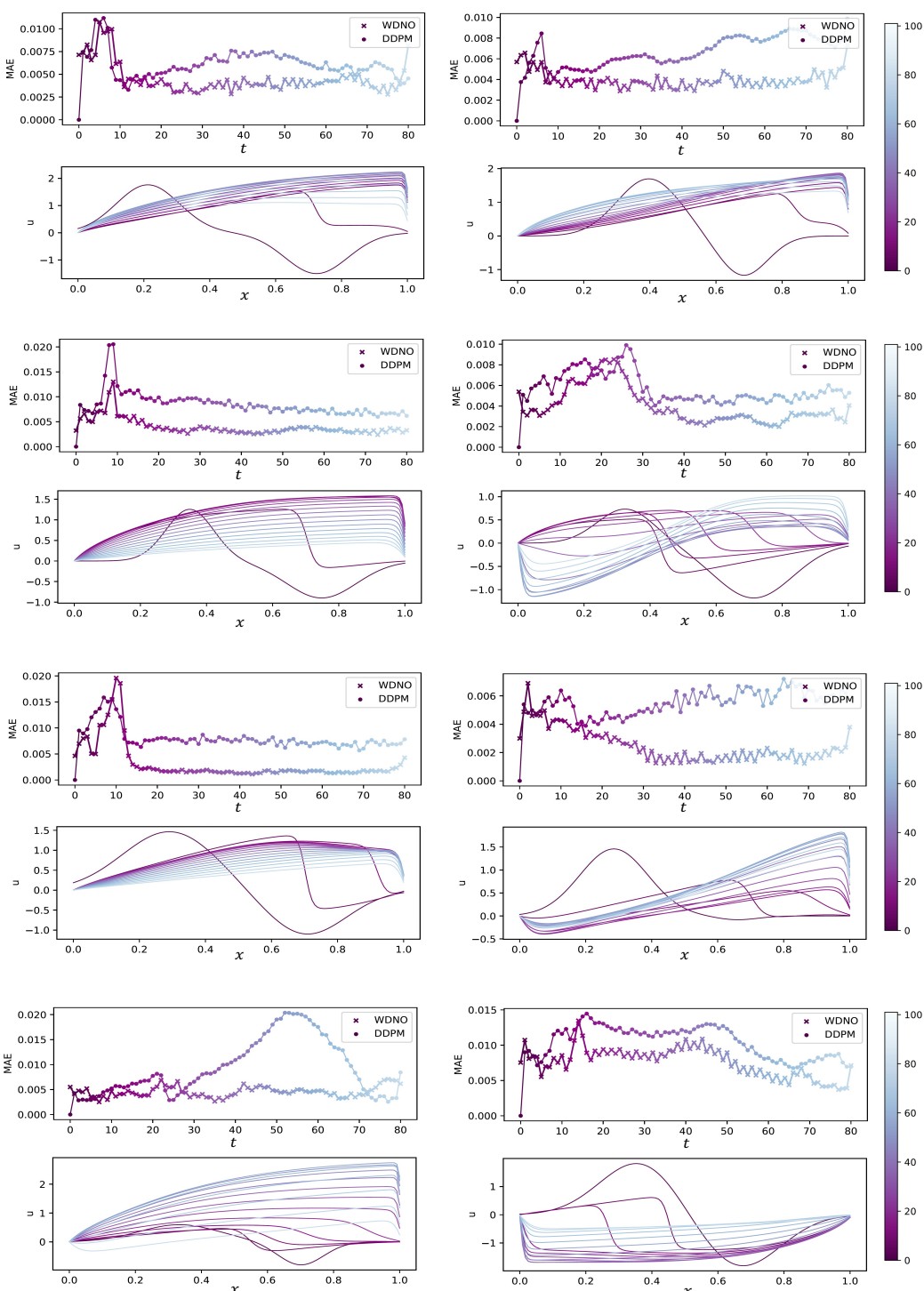

Figure 9: **Visualizations of WDNO's and DDPM's MAE on the 1D Burgers' equation.**

## C.3  APPROXIMATE SCALE INVARIANCE

We conduct experiments to verify approximate scale invariance by training FNOs on original, once-downsampled, twice-downsampled, and mixed datasets, then testing at these three resolutions. From Table 6, it is evident that the model trained on the mixed dataset performs better than those trained at specific resolutions.

Table 6: **Approximate scale invariance on the 1D Burgers' equation.**

|  | Original | Once-downsampled | Twice-downsampled |
|---|---|---|---|
| Mix | 4.72e-04 | 4.69e-04 | 4.95e-04 |
| Individual | 4.94e-04 | 5.42e-04 | 4.94e-04 |

## C.4  SENSITIVITY ANALYSIS

To conduct a sensitivity analysis of key hyper-parameters, we analyze the impact of wavelet type, guidance weight $\lambda$, DDIM sampling steps, and coefficient $\eta$ on 1D simulation and control. As shown in Table 7 and Table 8, WDNO is not sensitive to hyper-parameters.

Table 7: **Results of simulation on 1D Burgers' equation.**

(a) **DDIM step.**

|  | MSE |
|---|---|
| 20 | 0.00022 |
| 40 | 0.00017 |
| 50 | 0.00014 |
| 100 | 0.00015 |
| 200 | 0.00017 |

(b) **DDIM $\eta$.**

|  | MSE |
|---|---|
| 0.2 | 0.00020 |
| 0.5 | 0.00020 |
| 0.8 | 0.00017 |
| 1 | 0.00014 |

(c) **Wavelet type.**

|  | MSE |
|---|---|
| 0.2 | 0.00020 |
| 0.5 | 0.00020 |
| 0.8 | 0.00017 |
| 1 | 0.00014 |

Table 8: **Results of control on 2D incompressible fluid.**

(a) **DDIM step.**

|  | Results |
|---|---|
| 20 | 0.0223 |
| 40 | 0.0215 |
| 50 | 0.0205 |
| 100 | 0.0200 |
| 200 | 0.0217 |

(b) **DDIM $\eta$.**

|  | Results |
|---|---|
| 0.2 | 0.2285 |
| 0.5 | 0.0694 |
| 0.8 | 0.0244 |
| 1 | 0.0205 |

(c) **Guidance weight (1e4).**

|  | Results |
|---|---|
| 9 | 0.0213 |
| 10 | 0.0207 |
| 11.5 | 0.0205 |
| 12.5 | 0.0205 |
| 13 | 0.0215 |

## C.5  ROBUSTNESS

In addition, to test the robustness of WDNO, we first conduct 1D experiments with a $0.1$ probability of noise in the control sequence. Table 9 shows that WDNO outperforms other learning-based methods, showing its robustness. We also give the results (mean $\pm$ std) of 1D control averaged over $50$ testing samples in Table 10, showing that WDNO's std is relatively low.

Table 9: **1D control experiments with a 0.1 probability of noise in the control sequences.**

| Methods | $\mathcal{J}$ |
|---|---|
| PID (surrogate-solver) | 0.6644 |
| SAC (pseudo-online) | 0.2166 |
| SAC (offline) | 0.3979 |
| BC | 0.2457 |
| BPPO | 0.2392 |
| SL | 0.0348 |
| DDPM | 0.0701 |
| **WDNO (ours)** | 0.0305 |

Table 10: **Results of control tasks (mean±std).** Bold font denotes the best model and the runner-up is underlined.

| Method | Results |
|---|---|
| PID (surrogate) | $0.6645 \pm 0.5940$ |
| SAC (pseudo-online) | $0.1376 \pm 0.1729$ |
| SAC (offline) | $0.3210 \pm 0.2733$ |
| BC (surrogate) | $0.2998 \pm 0.1137$ |
| BPPO (surrogate) | $0.3075 \pm 0.1178$ |
| SL | $0.0235 \pm 0.0171$ |
| DDPM | $0.0272 \pm 0.0198$ |
| **WDNO (ours)** | $0.0205 \pm 0.0198$ |

C.6 COMPUTATIONAL RESOURCES

We provide inference times for a batch size of 1 on A100 in Table 11. It is evidence that WDNO's runtime is moderate, and its relatively large parameter count is due to the U-Net base model, which can be replaced with smaller models. In addition, we test WDNO's total training and inference times. As shown in Table 12, WDNO has reasonable spatial and temporal costs. Notably, for the 1D Burgers' equation, WDNO achieves the lowest training time, as shown in Table 13.

Table 11: **Number of parameters and inference time (s) results of 1D Burgers' equation.**

(a) **Control task.**

| Methods | Parameters | Time |
|---|---|---|
| PID (surrogate) | 4034952 | 0.081 |
| SAC (pseudo-online) | 89011116 | 0.503 |
| SAC (offline) | 88854770 | 0.466 |
| BC | 1543408 | 0.075 |
| BPPO | 6579171 | 0.079 |
| SL | 156346 | 181.35 |
| DDPM | 140703746 | 1.903 |
| **WDNO (ours)** | 140748553 | 1.131 |

(b) **Simulation task.**

| Methods | Parameters | Time |
|---|---|---|
| WNO | 153408 | 0.0105 |
| MWT | 2733978 | 0.0093 |
| OFormer | 2556321 | 0.0338 |
| FNO | 4769601 | 0.003 |
| CNN | 156346 | 0.277 |
| DDPM | 140703746 | 1.884 |
| **WDNO (ours)** | 140748553 | 0.966 |

Table 12: **Total training and inference time of WDNO.**

|  | Control | | Simulation | |
|---|---|---|---|---|
|  | Time | Space (MB) | Time | Space (MB) |
| 1D train (A100) | 2.4h | 7165 | 2.5h | 7165 |
| 1D inference (A100) | 455s | 3691 | 5.2s | 3761 |
| 2D train (2A100) | 7.8h | 8043+8039 | 7.9h | 8043+8039 |
| 2D inference (A100) | 2676s | 30255 | 70.9s | 16621 |

Table 13: **Training time (h) results of 1D Burgers' equation.**

| Methods | Time (h) |
|---|---|
| WNO | 4.5 |
| MWT | 6.5 |
| OFormer | 19.7 |
| FNO | 10.5 |
| CNN | 63.8 |
| DDPM | 7.8 |
| **WDNO (ours)** | 2.5 |

Moreover, in Table 14, we provide the time and space required for WDNO to generate a batch of size 5 during inference on the 1D Burgers' equation experiment, without super-resolution, and with one, two, and three levels of super-resolution. As shown in Table, with each increase in the level of resolution, the required time and space increase, and the growth rate is increasing. We can infer that as the level of super-resolution increases, the spatiotemporal costs also rise, indicating potential areas for further algorithm optimization.

Table 14: **Inference time and space of 1D super resolution.**

| Level of super resolution | 0 | 1 | 2 | 3 |
|---|---|---|---|---|
| Time (s) | 1.7 | 1.8 | 6.9 | 28.1 |
| Space (MB) | 1711 | 1831 | 3503 | 10631 |

## C.7 IMPORTANCE OF GUIDANCE

Since $\lambda$ in Eq. 4 is a hyperparameter, it can be set to zero, which means not including this term during denoising. In practice, we have selected the best-performing $\lambda$. To further show the effectiveness, we have also provided the 1D control results with and without this term in Table 15. We see that without this term, the performance drops a lot.

Table 15: **Results of control on the 1D Burgers' equation.**

| $\lambda$ | Results |
|---|---|
| 0 | 0.3360 |
| 120000 | 0.0205 |

## C.8 ZERO-SHOT SUPER-RESOLUTION

In Table 16 and Table 17, we provide results in Section 4.6, which reveal that our method is outstanding in zero-shot super-resolution.

Table 16: **Mean squared error of zero-shot super-resolution on the 1D Burgers' equation.**

| Methods | 0 times | 1 times | 2 times | 3 times |
|---|---|---|---|---|
| WNO (linear) | 0.0110 | 0.6284 | 1.4474 | 2.0588 |
| FNO (linear) | 0.00312 | 0.02463 | 0.04458 | 0.05610 |
| WDNO (linear) | 0.00259 | 0.00074 | 0.00036 | **0.00035** |
| WNO (nearest) | 0.0079 | 0.6491 | 1.5007 | 2.0588 |
| FNO (nearest) | 0.00439 | 0.02473 | 0.04450 | 0.05610 |
| WDNO (nearest) | 0.00382 | 0.00110 | 0.00046 | **0.00035** |

Table 17: **Mean squared error of zero-shot super-resolution on the 2D incompressible fluid.**

| Methods | 0 times | 1 times |
|---|---|---|
| FNO (linear) | 0.09497 | 0.01692 |
| WDNO (linear) | 0.09309 | **0.00765** |
| FNO (nearest) | 0.11963 | 0.01692 |
| WDNO (nearest) | 0.12002 | **0.00765** |

## D   RELATED WORK

**PDE simulation.** Solving a family of PDEs can be regarded as approximating nonlinear operators in the functional space, where neural operators have recently proved effective (Kovachki et al., 2023), such as DeepONet (Lu et al., 2019), FNO (Li et al., 2021). GNOT (Hao et al., 2023), LNPDE (Iakovlev et al., 2023a), CROM (Chen et al., 2022b), DINo (Yin et al., 2022), MagNet (Boussif et al., 2022), Transolver (Wu et al., 2024a) and CNO (Raonic et al., 2024). Additionally, scientific knowledge such as Clifford algebras (Brandstetter et al., 2022a) and Koopman theory (Wang et al., 2022) has been incorporated into neural networks to improve neural operators' performance. There are also neural ODE based approaches able to simulate PDE systems (Iakovlev et al., 2022; Lagemann et al., 2023). Among them, some models explicitly learn inside functional spaces, such as the Fourier domain (Li et al., 2021) and the wavelet domain (Gupta et al., 2021; Tripura & Chakraborty, 2022; Cheng et al., 2024). However, most existing works mainly focus on simulating physical systems, lacking physical system control problems. Our work proposes a wavelet diffusion neural operator that excels in simulating physical systems and can naturally manage both control and super-resolution simulation tasks.

**Super-resolution tasks.** Super-resolution tasks aim to reconstruct high-resolution data from low-resolution data. In recent years, many studies on physical system simulation have focused on super resolution tasks. Some methods transfer the learning of dynamics into function space, naturally enabling zero-shot super resolution capabilities (Li et al., 2021; Tripura & Chakraborty, 2022; Cao et al., 2024). Additionally, some studies attempt to incorporate physical information, such as equation forms, into the model's learning process to achieve super-resolution (Gao et al., 2021; Jangid et al., 2022; Zayats et al., 2022; Jangid et al., 2022). Other studies primarily achieve super-resolution by injecting high-resolution information into a super-resolution model (Esmaeilzadeh et al., 2020; Ren et al., 2023; Shu et al., 2023). Given the importance of the super-resolution task, we propose leveraging approximate scale invariance to enable diffusion models to achieve super-resolution capabilities.

**Wavelet transform.** The wavelet transform, a powerful tool for signal processing and analysis, is widely utilized in designing deep learning algorithms. Due to its ability to decompose signals into high-frequency and low-frequency components, it is used to enhance robustness (Li et al., 2020a), improve accuracy (Li et al., 2020a; Liu et al., 2019), enable super-resolution (Guo et al., 2017; Huang et al., 2017), and extract features (Wang et al., 2021), among other applications. Two closely related works (Hui et al., 2022; Guth et al., 2022) incorporate the wavelet transform into the diffusion model, but these works do not involve space-time multi-resolution correlations. Also, our paper focuses on different tasks and emphasizes the operator characteristics in PDE systems, such as mapping between infinite-dimensional function spaces.

**Long-term predictions.** Error accumulation is a common challenge for transient PDE predictions, and several methods have been proposed to address it. Some works suggest training prediction models over multiple steps rather than a single step to enhance robustness in multi-step predictions (Lusch et al., 2018; Brandstetter et al., 2022b). Techniques such as noise injection into training data and adversarial training are employed to improve the model's resilience to small disturbances (Sanchez-Gonzalez et al., 2020; Lippe et al., 2024), while other works use geometric manifold learning to identify the intrinsic dimensions of observed systems, enabling robust predictions of underlying dynamics (Chen et al., 2022a).

**PDE control.** For the task of controlling physical systems governed by PDEs, various deep learning-based techniques have been proposed (Feng et al., 2023; Zhu et al., 2021; Degrave et al., 2022). A prominent class of methods is supervised learning (SL) (Holl et al., 2020; Hwang et al., 2022) which optimizes control input via backpropagation through a neural surrogate model. Unlike these methods, our approach does not rely on auto-regressive surrogate models but instead learns both entire state trajectories and control sequences. Besides, deep reinforcement learning (DRL) has been applied to various physical problems such as drag reduction (Rabault et al., 2019; Elhawary, 2020; Feng et al., 2023; Wang et al., 2024), heat transfer (Beintema et al., 2020; Hachem et al., 2021), and swimming (Novati et al., 2017; Verma et al., 2018). These methods often implicitly incorporate physical information and make decisions sequentially. In contrast, our approach generates entire trajectories, facilitating trajectory-level optimization while embedding physical insights learned by models. Additionally, physics-informed neural networks (PINNs) (Raissi et al., 2019) have recently been used for control (Mowlavi & Nabi, 2023), but they require explicit formulations of PDE dynamics. In contrast, our method is data-driven and can address a broader spectrum of complex physical system control problems without knowledge of the explicit PDE dynamics.

**Diffusion models.** The diffusion model (Ho et al., 2020b) is proficient in learning high-dimensional distributions and has succeeded in image and text generation (Dhariwal & Nichol, 2021). It has also demonstrated remarkable ability, including strong modeling capabilities in complex and high-dimensional systems and temporal stability, in scientific or engineering problems such as robot control (Janner et al., 2022; Ajay et al., 2022), fluid prediction (Li et al., 2024; Kohl et al., 2024), weather forecasting (Price et al., 2023), 3D human motion generation (Vahdat et al., 2022), PDE simulation based on sparse observation (Huang et al., 2024) and PDE control (Wei et al., 2024; Wu et al., 2024b). Among them, ACDM (Kohl et al., 2024) is an autoregressive model for fluid simulation, which means both training and inference are performed sequentially. The model predicts the next $k$ steps $u_{[k,2k-1]}$ based on the previous $k$ steps $u_{[0,k-1]}$. Then, using $u_{[k,2k-1]}$, it predicts $u_{[2k,3k-1]}$, and this process continues iteratively until the entire trajectory $u_{[0,T-1]}$ of $T$ steps is generated. In contrast, our proposed method predicts the entire trajectory $u_{[0,T-1]}$ directly in a single inference step based on the given $k$ initial steps, significantly reducing computational overhead. Bisides, generalization across different resolutions and modeling states with abrupt changes remain challenging, and our WDNO proposes a promising direction to tackle the challenges. Many works have focused on improving the well-posedness of functional space diffusion generation (Pidstrigach et al., 2023; Hagemann et al., 2023; Lim et al., 2023). Previous works commonly choose the Fourier space as the functional space (Lim et al., 2023; Hagemann et al., 2023). Our method differs by using the wavelet transform since it is better at approximating the important abrupt changes.

# E  PSEUDOCODE

To help understand the entire algorithm, we provide the pseudocode of WDNO's training and inference in Algorithm 1, and the visualization of WDNO's entire training and inference on 1D Burgers' equation in Figure 10.

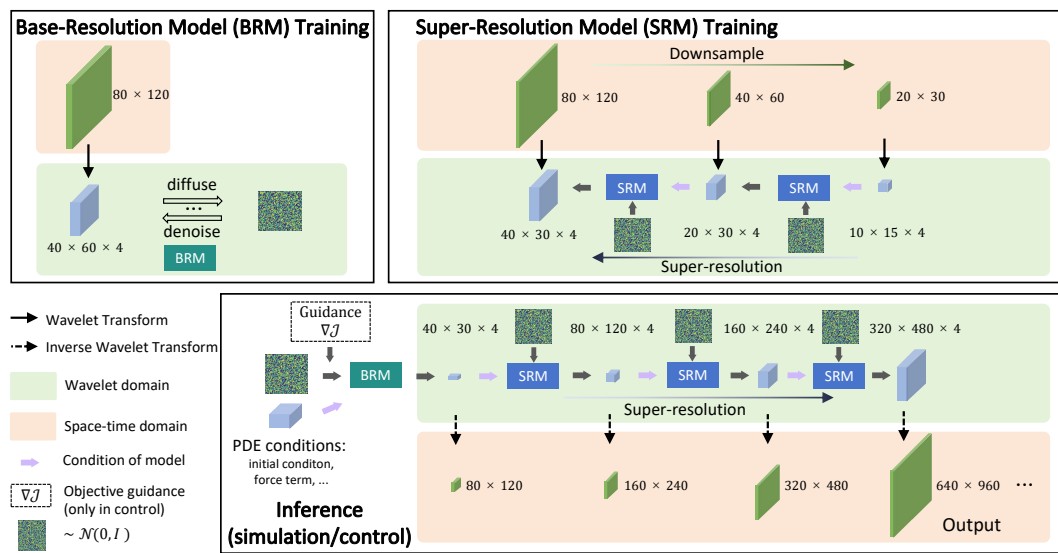

Figure 10: **Overview of WDNO**. The figure illustrates the training of Base-Resolution Model (BRM, top left), training of Super-Resolution Model (SRM, top right), and inference (bottom) of WDNO on 1D Burgers' equation. Through multi-resolution training and generation in wavelet space, WDNO is capable of generating superior simulation and control trajectories and conducting zero-shot super-resolution.

---

**Algorithm 1** Training and Sampling for WDNO

**Require** Diffusion models $\epsilon_\theta(W_{u_{[0,T]}}^{(k)}, W_a, k)$, objective $\mathcal{J}(\cdot)$ for control task, covariance matrix $\sigma^{2(k)}\mathbf{I}$, condition $W_a$, schedule $\bar\alpha_k$, hyperparameters $\lambda, \eta, K$

**Training:**
1: **repeat**
2:      $u_{[0,T]}^{(0)} \sim q(u_{[0,T]}^{(0)})$
3:      Apply the discrete wavelet transform to $u_{[0,T]}^{(0)}$ to get $W_{u_{[0,T]}}^{(0)}$
4:      $k \sim \text{Uniform}(1, \ldots, K)$
5:      $\epsilon \sim \mathcal{N}(0, \mathbf{I})$
6:      Take gradient descent step on $\nabla_\theta \|\epsilon - \epsilon_\theta(\sqrt{\bar\alpha_k} W_{u_{[0,T]}}^{(0)} + \sqrt{1 - \bar\alpha_k}\epsilon, k)\|^2$
7: **until** converged

**Sampling:**
1: $W_{u_{[0,T]}}^{(K)} \sim \mathcal{N}(0, \mathbf{I})$
2: **for** $k = K, \ldots, 1$ **do**
3:      $\boldsymbol{\xi} \sim \mathcal{N}(0, \mathbf{I})$ if $k > 1$, else $\boldsymbol{\xi} = 0$
4:      $W_{u_{[0,T]}}^{(k-1)} = W_{u_{[0,T]}}^{(k)} - \eta(\epsilon_\theta(W_{u_{[0,T]}}^{(k)}, W_a, k) + \boldsymbol{\xi}$ for simulation task
       $W_{u_{[0,T]}}^{(k-1)} = W_{u_{[0,T]}}^{(k)} - \eta(\epsilon_\theta(W_{u_{[0,T]}}^{(k)}, W_a, k) + \lambda\nabla\mathcal{J}(W_{u_{[0,T]}}^{(k)}) + \boldsymbol{\xi}$ for control task
5: **end for**
6: Apply the inverse discrete wavelet transform to $W_{u_{[0,T]}}^{(0)}$ to get $u_{[0,T]}^{(0)}$
7: **return** $u^* = u_{[0,T]}^{(0)}$

---

# F    ADDITIONAL DETAILS FOR 1D BURGERS' EQUATION CONTROL

## F.1    EXPERIMENT SETTING

The equation takes the form

$$
\begin{cases}
\frac{\partial u(t,x)}{\partial t} = -u(t,x) \cdot \frac{\partial u(t,x)}{\partial x} + \nu \frac{\partial^2 u(t,x)}{\partial x^2} + f(t,x) & \text{in } [0,T] \times D, \\
u(t,x) = 0 & \text{on } [0,T] \times \partial D, \\
u(0,x) = u_0(x) & \text{at } \{t=0\},
\end{cases}
\tag{8}
$$

where $u_0$ is the initial condition, the diffusion coefficient $\nu = 0.01$, $T = 8$ and $D = [0,1]$.

During inference, alongside the control sequence $f(t,x)$, our diffusion model generates states $\mu(t,x)$. Besides, some models produce surrogate states $\mu(t,x)$ when fed with the control $f(t,x)$. However, the state deviation $\int_D |u(T,x) - u^*(x)| dx$ in our reported evaluation metric $\mathcal{J}$ is always based on the output $u(T,x) = u_{\text{g.t.}}(T,x)$ of the ground-truth solver given the control force $f(t,x)$.

The solver solves the Burgers' equation (Eq. 8) as described in Appendix F.2, where the internal grid size of the ground-truth numerical solver is consistently at high-resolution ($[80 \times 16, 120 \times 16]$). When model outputs are at a lower resolution, we linearly interpolate them before feeding them into the solver.

## F.2    DATA GENERATION

We use the finite difference method (referred to as the solver or ground-truth solver henceforth) to solve the Burgers' equation of Eq. 8 and generate the training data for the 1D Burgers' equation. Specifically, the initial state $u_0(x)$ and the control force $f(t,x)$ are both randomly generated, and then the state's evolution $u(t,x)$ is numerically simulated using the solver. In the numerical simulation using the ground-truth solver, a domain of $x = [0,1]$, $t = [0,8]$ is simulated. The space is discretized into $120 \times 16$ grids and time discretized into $4800 \times 16$ steps. However, only 80 time stamps are stored in the dataset, and the control sequence $f$ is kept constant between two time stamps.

After simulation, we downsample by 16 times both spatially and temporally before saving the dataset. Therefore, the data size of each trajectory is $[81, 120]$ for the state $u$ and $[80, 120]$ for the force $f$. As for the super-resolution dataset for super-resolution simulation, we downsample the original data with shapes $[80 \times N + 1, 120 \times N]$ for $u$ and $[80 \times N, 120 \times N]$ where $N = 2, 3, 4$ corresponding to $1, 2, 3$ times super-resolution in Table 16.

The initial value $u(0,x)$ is a superposition of two Gaussian functions $u(0,x) = \sum_{i=1}^{2} a_i e^{-\frac{(x-b_i)^2}{2\sigma_i^2}}$, where $a_i, b_i, \sigma_i$ are all randomly sampled from uniform distributions: $a_1 \sim U(0,2)$, $a_2 \sim U(-2,0)$, $b_1 \sim U(0.2,0.4)$, $b_2 \sim U(0.6,0.8)$, $\sigma_1 \sim U(0.05,0.15)$, $\sigma_2 \sim U(0.05,0.15)$. Similarly, the control sequence $f(x,t)$ is also a superposition of 8 Gaussian functions $f(t,x) = \sum_{i=1}^{8} a_i e^{-\frac{(x-b_{1,i})^2}{2\sigma_{1,i}^2}} e^{-\frac{(t-b_{2,i})^2}{2\sigma_{2,i}^2}}$, where each parameter is independently generated as follows: $b_{1,i} \sim U(0,1)$, $b_{2,i} \sim U(0,1)$, $\sigma_{1,i} \sim U(0.1,0.4)$, $\sigma_{2,i} \sim U(0.1,0.4)$, while $a_1 \sim U(-1.5,1.5)$ and for $i \geq 2$, $a_i \sim U(-1.5,1.5)$ or 0 with equal probabilities. $u(t,x)$, $(t \neq 0)$ is then numerically simulated (using the ground-truth solver) given $u(0,x)$ and $f(t,x)$ based on Eq. 8. The dataset generation setting is based on previous works Hwang et al. (2022); Wei et al. (2024).

we generate 40000 trajectories for the training set. For the Burgers' equation control task, we generate another 50 trajectories for testing. For the super-resolution task, we generate another 2000 trajectories for testing in the $0\times$ super-resolution task (which is the original resolution). In $1,2,3\times$ super-resolution tasks, another 100 samples are generated and shared across the three different super-resolution settings.

## F.3    DATA PREPARATION FOR WDNO

We perform a 2D wavelet transform on the original data using the *bior2.4* wavelet basis and the 'periodization' mode, implemented using the `pytorch_wavelets` package (Cotter, 2019). This transform the data, originally sized $81 \times 120$, into four sets of wavelet coefficients, each sized $41 \times 60$.

Among these four sets of coefficients, there is one set of coarse coefficients and three sets of detail coefficients. For the multi-resolution dataset used to train the Super-Resolution Model, obtained through downsampling, we conduct wavelet transforms on data sizes $41 \times 60$, $21 \times 31$, and $11 \times 15$, which correspond to four sets of wavelet coefficients sized $21 \times 30$, $11 \times 15$, and $6 \times 8$ respectively.

Notably, since the initial condition and the target state are 1D, we take the 1D wavelet transform, repeat the coefficients, and then concatenate them to other data.

When aligning the sizes of low-resolution wavelet coefficients with high-resolution ones by duplication, special handling is required at the boundaries due to the presence of odd numbers. Specifically, we duplicate the last temporal dimension of the high-resolution data once more to ensure that the sizes match perfectly.

### F.4  MODEL

The model architecture in this experiment follows the Denoising Diffusion Probabilistic Model (DDPM) (Ho et al., 2020b). In simulations, the Base-Resolution Model conditions on $u_0$ and $f$ to predict $u_{0,T}$. For control tasks, we condition on $u_0$, $u_T$ and apply guidance related to $\mathcal{J}$ to generate $f_{[0,T]}$. The Super-Resolution Model conditions the same variables as the Base-Resolution Model, with the addition of conditioning on the low-resolution data. Besides, to make the generation meet the initial condition and target state better, we involve the loss between the inverse wavelet transform of the initial condition and the target state's wavelet coefficient channel and the ground truth into the guidance. The hyperparameters on WDNO are recorded in Table 18.

Table 18: **Hyperparameters of the UNet architecture and training for the results of 1D Burgers' equation in Table 2a and Table 1**.

| Hyperparameter name | Base-Resolution Model | Super-Resolution Model |
|---|---|---|
| UNet $\epsilon_\phi(f)$ | | |
| Initial dimension | 128 | 128 |
| Downsampling/Upsampling layers | 4 | 4 |
| Convolution kernel size | 3 | 3 |
| Dimension multiplier | $[1, 2, 4, 8]$ | $[1, 2, 4, 8]$ |
| Resnet block groups | 8 | 8 |
| Attention hidden dimension | 32 | 32 |
| Attention heads | 4 | 4 |
| UNet $\epsilon_\theta(u, f)$ | | |
| Initial dimension | 128 | 128 |
| Downsampling/Upsampling layers | 4 | 4 |
| Convolution kernel size | 3 | 3 |
| Dimension multiplier | $[1, 2, 4, 8]$ | $[1, 2, 4, 8]$ |
| Resnet block groups | 8 | 8 |
| Attention hidden dimension | 32 | 32 |
| Attention heads | 4 | 4 |
| Training | | |
| Training batch size | 16 | 16 |
| Optimizer | Adam | Adam |
| Learning rate | 1e-4 | 1e-4 |
| Training steps | 190000 | 290000 |
| Learning rate scheduler | cosine annealing | cosine annealing |
| Inference | | |
| DDIM sampling iterations | 50 | 50 |
| $\eta$ of DDIM Sampling | 1 | 1 |
| Intensity of guidance in control | 120000 | 0 |
| Scheduler of guidance | cosine | cosine |

# G  ADDITIONAL DETAILS FOR 1D COMPRESSIBLE NAVIER-STOKES EQUATION

## G.1  EXPERIMENT SETTING

This fluid dynamic equation takes the form

$$
\begin{cases}
\partial_t \rho + \nabla \cdot (\rho \mathbf{v}) = 0, \\
\rho \left( \partial_t \mathbf{v} + \mathbf{v} \cdot \nabla \mathbf{v} \right) = -\nabla p + \eta \Delta \mathbf{v} + \left( \zeta + \frac{\eta}{3} \right) \nabla (\nabla \cdot \mathbf{v}), \\
\partial_t \left( \epsilon + \frac{\rho v^2}{2} \right) + \nabla \cdot \left[ \left( \epsilon + p + \frac{\rho v^2}{2} \right) \mathbf{v} - \mathbf{v} \cdot \sigma' \right] = 0,
\end{cases}
\tag{9}
$$

where $\rho$ is the density, $\mathbf{v}$ is the velocity, $p$ is the pressure, $\epsilon = p/(\Gamma - 1)$ is the internal energy with $\Gamma = 5/3$, $\sigma'$ is the viscous stress tensor, and $\eta, \zeta$ are the shear and bulk viscosity, respectively. The sound velocity is defined as:

$$
c_s = \sqrt{\Gamma \frac{p}{\rho}},
\tag{10}
$$

and the Mach number is:

$$
M = \frac{|\mathbf{v}|}{c_s}.
\tag{11}
$$

The velocity field for turbulence is initialized as:

$$
\mathbf{v}(x, t = 0) = \sum_{i=1}^{n} A_i \sin(k_i x + \phi_i),
\tag{12}
$$

where $A_i = \frac{\bar{v}}{|k|^d}$, $d = 1, 2$ for 2D and 3D cases, and $\bar{v} = c_s M$.

The shock-tube field is initialized as:

$$
Q(x, t = 0) = (Q_L, Q_R),
\tag{13}
$$

where $Q = (\rho, \mathbf{v}, p)$, with random constants $Q_L$ and $Q_R$.

In detail, we use a 1D compressible Navier-Stokes equation dataset provided by `PDEBench`. The initial conditions include random fields, turbulent fields, and shock-tube fields. The random and turbulence fields are prepared by adding perturbations to a uniform background, while the shock-tube setup consists of piecewise constant values generating shocks and rarefactions. Boundary conditions allow waves to exit the domain, and numerical solutions are computed using second-order HLLC and central difference schemes. And we choose the most challenging dataset `'1D_CFD_Shock_Eta1.e-8_Zeta1.e-8_trans_Train.hdf5'`

Since the original data is of quite high resolution, we downsample it and the final resolution of the used data is $81 \times 120$, the same as the 1D Burgers' equation.

## G.2  DATA PREPARATION FOR WDNO

Since the data size is the same as Appendix F, the data preparation is almost the same. We also perform a 2D wavelet transform on the original data using the *bior2.4* wavelet basis and the 'periodization' mode, implemented using the `pytorch_wavelets` package (Cotter, 2019). As for the initial condition, we take the 1D wavelet transform repeat the coefficients, and then concatenate them to other data.

## G.3  MODEL

The model architecture in this experiment follows Appendix F. The hyperparameters on WDNO are recorded in Table 19.

Table 19: **Hyperparameters of the UNet architecture and training for the results of 1D compressible Navier-Stokes equation in Table 1.**

| Hyperparameter name | Base-Resolution Model |
|---|---|
| UNet $\epsilon_\phi(f)$ | |
| Initial dimension | 128 |
| Downsampling/Upsampling layers | 4 |
| Convolution kernel size | 3 |
| Dimension multiplier | $[1, 2, 4, 8]$ |
| Resnet block groups | 8 |
| Attention hidden dimension | 32 |
| Attention heads | 4 |
| UNet $\epsilon_\theta(u, f)$ | |
| Initial dimension | 128 |
| Downsampling/Upsampling layers | 4 |
| Convolution kernel size | 3 |
| Dimension multiplier | $[1, 2, 4, 8]$ |
| Resnet block groups | 8 |
| Attention hidden dimension | 32 |
| Attention heads | 4 |
| Training | |
| Training batch size | 16 |
| Optimizer | Adam |
| Learning rate | 1e-4 |
| Training steps | 190000 |
| Learning rate scheduler | cosine annealing |
| Inference | |
| DDIM sampling iterations | 850 |
| $\eta$ of DDIM Sampling | 1 |
| Scheduler of guidance | cosine |

# H  ADDITIONAL DETAILS FOR 2D INCOMPRESSIBLE FLUID

## H.1  EXPERIMENT SETTING

The equation takes the form

$$\begin{cases} \frac{\partial \mathbf{v}(t,x)}{\partial t} + \mathbf{v}(t,x) \cdot \nabla \mathbf{v}(t,x) - \nu \nabla^2 \mathbf{v}(t,x) + \nabla p(t,x) = f(t,x), \\ \nabla \cdot \mathbf{v}(t,x) = 0, \\ \mathbf{v}(0,x) = \mathbf{v}_0(x), \end{cases} \tag{14}$$

where $f$ is the external force, $p$ denotes pressure, $\mathbf{v}$ is the velocity and $\nu$ is the viscosity coefficient.

## H.2  DATA PREPARATION FOR WDNO

We performed a 3D wavelet transform on the original data using the *bior1.3* wavelet basis and 'zero' mode, implemented through the `Pytorch Wavelet Toolbox (ptwt)` (Wolter et al., 2024). The data of size $32 \times 64 \times 64$ are transformed into eight sets of wavelet coefficients, each sized $18 \times 34 \times 34$. Among these, there is one set of coarse coefficients and seven sets of detail coefficients. For the multi-resolution dataset used to train the Super-Resolution Model, obtained through downsampling, we do wavelet transforms on data sizes $32 \times 32 \times 32$ and $32 \times 16 \times 16$ corresponding to wavelet coefficient with sizes $18 \times 18 \times 18$, $18 \times 10 \times 10$ respectively.

Specifically, since the initial condition and the percentage of smoke through the target bucket are 2D and 1D respectively, we take the 2D and 1D wavelet transform and repeat the coefficients to concatenate them.

Similarly, to align the duplicated low-resolution data with the high-resolution data, we duplicate the boundary values on both sides of the dimensions requiring super-resolution in the high-resolution data.

## H.3 MODEL

In this paper, the architecture of the three-dimensional U-net we employ is inspired by the previous work (Ho et al., 2022). In this experiment, we use spatial-temporal 3D convolutions. Specifically, there are three main modules in our U-net: a downsampling encoder, a middle module, and an upsampling decoder.

In the simulation, the diffusion model conditions the initial density and control to generate the entire trajectories of density, velocity, and percentage of smoke passing the target bucket. In the control problem, the diffusion model conditions on the initial density and takes the negative percentage of smoke through the target bucket at the last time step as the guidance. Same as the 1D case, to satisfy the initial condition better, the guidance also involves loss between the inverse wavelet transform of the initial condition's wavelet coefficient channel and the ground truth initial condition. The hyperparameters of the 3D-Unet architecture are in the Table 20.

Table 20: **Hyperparameters of 3D-Unet architecture in 2D experiments**.

| Hyperparameter Name | Value |
|---|---|
| Number of attention heads | 4 |
| Kernel size of conv3d | (3, 3, 3) |
| Padding of conv3d | (1,1,1) |
| Stride of conv3d | (1,1,1) |
| Kernel size of downsampling | (1, 4, 4) |
| Padding of downsampling | (1, 2, 2) |
| Stride of downsampling | (0, 1, 1) |
| Kernel size of upsampling | (1, 4, 4) |
| Padding of upsampling | (1, 2, 2) |
| Stride of upsampling | (0, 1, 1) |
| DDIM sampling iterations | 100 |
| $\eta$ of DDIM Sampling | 1 |
| Intensity of guidance in control | 100 |

## I  1D CONTROL BASELINES

### I.1  PID

Propercentageal Integral Derivative (PID) control (Li et al., 2006) is a versatile and effective method widely employed in numerous control scenarios. It operates by using the error, i.e., the difference between the desired target and the current state of a system. Due to its simplicity and effectiveness, PID control is often the default choice for many control problems. However, despite its widespread use, PID control faces challenges such as parameter adaptation and limitations in Single Input Single Output (SISO) systems.

In our study, the 1D Burgers' Equation Control problem presents a Multiple Input Multiple Output (MIMO) scenario, rendering direct application of PID control infeasible. Inspired by early works (Slama et al., 2019; Ding et al., 2022) that employed neural networks as PID parameter adapters, we integrated deep learning with PID control to address the MIMO control problem. As illustrated in Figure 11, the ANN (artificial neural network) PID uses a neural network to adapt PID parameters, enabling multiple sets of SISO PID control.



Figure 11: **The architecture of ANN PID Controller.** To use the MIMO PID controller to control state $u_t$ to target state $u_d$, we train a neural-network-based PID parameter planner to output MIMO PID parameters based on $Err_t$, then use the PID controller to output the control sequence $f_t$.

The neural network generating the PID parameters consists of two 1D convolutional layers, two fully connected layers, and four activation layers. We utilize the $L1$ loss between the current and target states as the training loss, and the Adam optimizer (Kingma & Ba, 2014) to train the model. Detailed architecture information is provided in Table 21.

Table 21: **Hyperparameters of network architecture and training for ANN PID**.

| Hyperparameter name | Full observation |
|---|---|
| Kernel size of conv1d | 3 |
| Padding of conv1d | 1 |
| Stride of conv1d | 1 |
| Activation function | Softsign |
| Batch size | 16 |
| Optimizer | Adam |
| Learning rate | 0.0001 |
| Loss function | MAE |

Given that PID is inherently a SISO control method, the ANN PID employs a neural network to derive multiple PID parameter sets, facilitating multiple SISO PID controls for MIMO control in the context of the Burgers' equation. However, ANN PID requires that the input and output dimensions match, thus it can only address problems with full observation and full control, or partial observation and partial control.

Additionally, the ANN PID controller has two training setups: one involving direct interaction with the solver, and the other involving interaction with the 1D surrogate model.

## I.2 SAC

The Soft Actor-Critic (SAC) algorithm, developed by Haarnoja et al. (2018), represents a significant advancement in reinforcement learning techniques. Designed as an enhancement of the conventional Actor-Critic frameworks, SAC sets itself apart by incorporating an entropy regularization term in its loss function. This addition promotes more effective exploration by simultaneously maximizing the expected cumulative reward and the entropy of the policy itself, leading to improved decision-making processes in complex environments.

Compared to the Deep Deterministic Policy Gradient (DDPG) algorithm (Lillicrap et al., 2015; Pan et al., 2018), the Soft Actor-Critic (SAC) algorithm introduces entropy regularization that promotes more effective exploration and avoids premature convergence to suboptimal policies, a common drawback of DDPG's deterministic nature. Furthermore, SAC's twin Q-networks counteract the overestimation bias that can affect DDPG's value updates, resulting in more stable learning processes. The automatic adjustment of the temperature parameter in SAC also eases the balancing act between exploration and exploitation, minimizing the necessity for careful hyperparameter tuning. As a result, these features make SAC generally more sample-efficient and robust, especially in complex and continuous action spaces.

During training, experiences are stored in a replay buffer and sampled randomly to update the networks. Initially, the entire training set is loaded into the replay buffer. For offline SAC, this replay buffer remains unchanged. In contrast, online SAC alternates between gathering experiences through environment interaction and updating the networks using the replay buffer. Offline SAC, however, utilizes a surrogate model trained on the training set to gather experiences instead of the real environment. The policy network is optimized to maximize the expected return by considering both the Q-value and the entropy term. The critic networks are trained to minimize the error between their Q-value predictions and the target Q-values. To further stabilize training, SAC employs a target critic network, which is slowly updated with the weights from the main critic network. For inference, SAC uses the policy network to select the action with the highest probability.

To effectively guide the system towards the target state with accuracy and speed, it is essential to incorporate the distance between the state at each time step and the target state into the reward function. Therefore, the reward function for a given time step $t$, state $u_t$, target state $u_T$, and action $w_t$ is defined as follows:

$$r(t, u_t, y_T, w_t) = -\int_\Omega |u_t - u_d|^2 \ dx - \alpha \int_\Omega |w_t|^2 \ dx, \tag{15}$$

where $\Omega$ is the space domain and $\alpha$ is the weight of energy. We take the Adam optimizer (Kingma & Ba, 2014) to train the networks and update the temperature parameter. The detailed values of hyperparameters are provided in Table 22.

Table 22: **Hyperparameters of 1D SAC.**

| Hyperparameter name | Value |
| --- | --- |
| Hyperparameters for 1D Burgers' equation control: | |
| Discount factor for reward | 0.5 |
| Target smoothing coefficient | 0.05 |
| Learning rate of critic loss | 0.0003 |
| Learning rate of entropy loss | 0.003 |
| Learning rate of policy loss | 0.003 |
| Training batch size | 8192 |
| Number of episodes | 1500 |
| Number of model updates per simulator step | 50 |
| Value target updates per step | 15 |
| Size of replay buffer | 1000000 |
| weight of energy cost | 0.00002 |
| Number of trajectories interacted with the environment per step | 1 |
| Number of layers of critic networks | 3 |
| Number of hidden dimensions of critic networks | 4096 |
| Number of layers of the policy network | 5 |
| Number of hidden dimensions of the policy network | 4096 |
| Activation function | ReLU |
| Clipping's range of policy network's standard deviation output | $[e^{-20}, e^2]$ |

### I.3 SUPERVISED LEARNING

The paper (Hwang et al., 2022) proposes a supervised-learning-based control algorithm that takes a neural operator as a surrogate model to solve control problems. It contains two stages. In the first stage, we take a neural operator to learn the PDE constraint as Hwang et al. (2022). Two VAEs based on CNN learn to project state $u$, control signal $f$ into the latent space, and a CNN learns the transition from $u_t$ to $u_{t+1}$ in the latent space. In the second stage, these three neural networks are used as surrogate models to calculate the gradient of the objective function with respect to the control input and optimize the control signal $f$.

During optimization, the reconstruction loss for the control force is also included to guide it out of the adversarial mode of the surrogate model. We consider the control $f$ as a learnable parameter and update it with the LBFGS optimizer (Liu & Nocedal, 1989). The hyperparameters of this supervised learning method are recorded in Table 23.

Table 23: **Hyperparameters of inference the 1D supervised learning method**.

| Hyperparameter name | Value |
| --- | --- |
| Hyperparameters for 1D Burgers' equation control | |
| Learning rate of $w$ updating | 0.1 |
| Number of epochs | 100 |
| Weight of average objective function loss | 1 |
| Weight of average reconstruction loss | 0.01 |
| Termination tolerance on first-order optimality of LBFGS optimizer | 1e-5 |
| Termination tolerance on parameter changes LBFGS optimizer | 1e-5 |

## I.4 BPPO

The Behavior Proximal Policy Optimization (BPPO) algorithm, introduced by (Zhuang et al., 2023), is an advanced reinforcement learning method that combines the strengths of Proximal Policy Optimization (PPO) with elements of behavior cloning. BPPO is an offline algorithm designed to monotonically improve the behavior policy in a manner akin to PPO. Due to the inherent conservatism of PPO, BPPO restricts the ratio of the learned policy to the behavior policy within a specific range, similar to other offline RL methods, which ensures the learned policy closely aligns with the behavior policy. By leveraging the conservatism of online on-policy algorithms, BPPO effectively addresses the overestimation issue often encountered in offline RL settings.

The algorithm begins by estimating a behavior policy using behavior cloning and then iteratively improves a target policy using the PPO objective with a behavior constraint. This process of policy improvement, advantage estimation, and policy update enables BPPO to refine the target policy while ensuring it remains close to the behavior policy. By integrating the strengths of online on-policy methods with tailored offline RL techniques, BPPO has demonstrated promising results on the D4RL benchmark, surpassing state-of-the-art offline RL algorithms.

During training, BPPO first initializes the behavior policy $\pi_\beta$ and the target policy $\pi_\theta$. The behavior policy $\pi_\beta$ is then estimated using behavior cloning to replicate the behavior demonstrated in the offline dataset. Subsequently, the target policy $\pi_\theta$ is optimized using the PPO objective with a behavior constraint, ensuring the target policy remains close to the behavior policy. The advantage function $A^{\pi_\beta}$ is then estimated using the behavior policy $\pi_\beta$ to evaluate the quality of actions taken by the target policy. Finally, the target policy is updated by maximizing the PPO objective with the estimated advantage function, and adjusting the policy parameters to enhance performance. In the implementation, a state value network and a Q value network are pre-trained using the state, action, and reward data from the offline dataset.

In practice, to enable the system to approximate the target state accurately and swiftly, it is essential to incorporate the distance between the state at each time step and the target state into the reward function. Thus, the reward function at time step $t$, given the state $u_t$, target state $u_d$, and action $f_t$, is defined as follows:

$$r(t, u_t, u_d, f_t) = -\int_\Omega |u_t - u_d|^2 \mathrm{d}x - \alpha \int_\Omega |f_t|^2 \mathrm{d}x,$$

where $\Omega$ is the space domain and $\alpha$ is the weight of energy. We use the Adam optimizer (Kingma & Ba, 2014) to train the networks and update the temperature parameter. The specific values of the hyperparameters used are detailed in Table 24.

Table 24: **Hyperparameters of 1D BPPO.**

| Hyperparameter name | Value |
|---|---|
| Hyperparameters for 1D Burgers' equation control: | |
| State value network: | |
| Learning rate of value network | $1 \times 10^{-4}$ |
| Steps of value network | $2 \times 10^{6}$ |
| Number of layers of value network | 3 |
| Batch size of value network | 512 |
| Number of hidden dimensions of value network | 512 |
| Q value network: | |
| Learning rate of Q network | $1 \times 10^{-4}$ |
| Steps of Q network | $2 \times 10^{6}$ |
| Number of layers of Q network | 2 |
| Batch size of Q network | 512 |
| Number of hidden dimensions of Q network | 1024 |
| Target Q network updates per step | 2 |
| Soft update factor | 0.005 |
| Discount factor for reward | 0.99 |
| Behavior cloning: | |
| Learning rate of BC | $1 \times 10^{-4}$ |
| Training batch size of BC | 512 |
| Number of episodes of BC | $5 \times 10^{5}$ |
| BPPO: | |
| Number of episodes of BPPO | $1 \times 10^{2}$ |
| Number of layers of policy networks | 2 |
| Number of hidden dimensions of policy networks | 1024 |
| Learning rate of BPPO | $1 \times 10^{-5}$ |
| Training batch size of BPPO | 512 |
| Clip ratio of BPPO | 0.25 |
| Weight decay factor | 0.96 |
| Weight of advantage function | 0.9 |
| Size of replay buffer | $2 \times 10^{6}$ |
| Activation function | ReLU |

Table 25: **Hyperparameters of 1D BC.**

| Hyperparameter name | Value |
|---|---|
| Hyperparameters for 1D Burgers' equation control: | |
| Learning rate | $1 \times 10^{-4}$ |
| Training batch size | 512 |
| Number of episodes | $5 \times 10^{5}$ |
| Size of replay buffer | $2 \times 10^{6}$ |
| Number of layers of policy networks | 2 |
| Number of hidden dimensions of policy networks | 1024 |
| Activation function | ReLU |

## I.5  BC

The Behavior Cloning (BC) algorithm, introduced by (Pomerleau, 1988), is a foundational technique in imitation learning. BC is designed to derive policies directly from expert demonstrations, utilizing supervised learning to associate states with corresponding actions. This method eliminates the necessity for exploratory steps commonly required in reinforcement learning by replicating the actions observed in expert demonstrations. One of the significant advantages of BC is that it does not

involve interacting with the environment during the training phase, which streamlines the learning process and diminishes the demand for computational resources.

In this approach, a policy network is trained using standard supervised learning strategies aimed at reducing the discrepancy between the actions predicted by the model and those performed by the expert in the dataset. The commonly used loss function for this purpose is the mean squared error between the predicted actions and expert actions. The dataset for training comprises state-action pairs harvested from these expert demonstrations. During inference, the model is evaluated using the same objective function as used in SAC. The specific hyperparameters utilized are detailed in Table 25.

## J  1D SIMULATION BASELINES

### J.1  FNO

FNO represents a deep learning framework capable of mapping between infinite-dimensional spaces. By parameterizing the integral kernel in Fourier space, FNO processes input through a sequence of Fourier layers, performing linear transformations in the Fourier domain for efficient convolutions. This architecture supports zero-shot super-resolution, allowing models trained on lower resolutions to predict at higher resolutions without retraining.

In 1D experiments, we train the FNO model using the initial state and all controls as the input and using the rest states as the output. The parameters are outlined in Table 26.

Table 26: **Hyperparameters of 1D FNO.**

| Hyperparameter name | Value |
|---|---|
| Hyperparameters for 2D Burgers' equation: | |
| Number of modes to keep in Fourier Layer | 16 |
| Width of the FNO (*i.e.* number of channels) | 64 |
| Number of input channel | 3 |
| Number of output channel | 1 |
| Number of hidden channels of the lifting block | 256 |
| Number of hidden channels of the projection block | 256 |
| Number of Fourier Layers | 4 |
| Expansion parameter of MLP layer | 0.5 |
| Non-Linearity module | Gelu |
| Rank of the tensor factorization of the Fourier weights | 1.0 |
| Mode of domain padding | one-sided |
| Learning rate | $1 \times 10^{-4}$ |
| Optimizer | Adam |
| Training epochs | 1000 |
| Learning rate scheduler | Cosine |
| Training batch size | 50 |

### J.2  WNO

The Wavelet Neural Operator (WNO) (Tripura & Chakraborty, 2022) is a novel operator learning algorithm that blends integral kernel with wavelet transformation. We record the hyperparameters of it in 1D simulation in Table 27. On 1D Burgers' equation and 1D compressible Navier-Stokes equation, we choose 'sym4', 'bior2.4' wavelet respectively.

Table 27: **Hyperparameters of 1D WNO**.

| Hyperparameter name | Value |
|---|---|
| Hyperparameters of the model architecture | |
| Type of wavelet | sym4 |
| Level of wavelet decomposition | 5 |
| Uplifting dimension | 40 |
| Number of wavelet layers | 4 |
| Training | |
| Training batch size | 100 |
| Optimizer | Adam |
| Learning rate | 1e-3 |
| Training epochs | 1000 |
| Learning rate scheduler | StepLR |

## J.3 CNN

Convolutional Neural Network is the key block in deep learning. For 1D simulation, our CNN model is based on Hwang et al. (2022). Details of model architecture and training can be found in Table 28

Table 28: **Hyperparameters of 1D CNN**.

| Hyperparameter name | Value |
|---|---|
| Autoencoder of state | |
| Convolution kernel size | 5 |
| Convolution padding | 2 |
| Activation function | ELU |
| Latent vector size | 256 |
| Autoencoder of force | |
| Convolution kernel size | 5 |
| Convolution padding | 2 |
| Activation function | ELU |
| Latent vector size | 256 |
| Training | |
| Training batch size | 5100 |
| Optimizer | Adam |
| Learning rate | 1e-3 |
| Training epochs | 500 |
| Learning rate scheduler | cosine annealing |

## J.4 MWT

To compare with other wavelet-based methods, we mainly implement the 1D baseline adapted from Gupta et al. (2021). We select 'legendre' wavelet here, following the original work. More configurations can be found in Table 29.

Table 29: **Configuration of 1D MWT**.

| Hyperparameter name | Value |
|---|---|
| Wavelet basis | legendre |
| Number of Fourier modes | 10 |
| Kernel size | 4 |
| Training batch size | 256 |
| Training epochs | 300 |
| Optimizer | Adam |
| Learning rate scheduler | MultiStepLR |

## J.5 OFORMER

The Operator Transformer (OFormer)Li et al. (2023) is a novel framework built upon self-attention, cross-attention, and a set of point-wise multilayer perceptrons. Details of model architecture and training can be found in Table 30.

Table 30: **Hyperparameters of 1D OFormer.**

| Hyperparameter name | Value |
|---|---|
| Encoder | |
| Type | SpatialEncoder2D |
| Input Channels | 3 |
| Embedding Dim of Token | 96 |
| Embedding Dim of Encoded Sequence | 256 |
| Heads | 4 |
| Depth | 6 |
| Resolution | 120 |
| Dropout of Embedding | 0.05 |
| Decoder | |
| Type | PointWiseDecoder2DSimple |
| Latent Channels | 256 |
| Out Channels | 1 |
| Scale | 0.5 |
| Res | 120 |
| Training | |
| Training batch size | 32 |
| Iteration | 50000 |
| Learning rate | 1e-4 |

## J.6 LADID

In the dynamics trajectory prediction community, predicting trajectories of dynamical systems is of interest (Chen et al., 2018). Among them, MS-L-NODE (Iakovlev et al., 2023b) is a representative method that is dedicated to learning the system invariant dynamics. Since MS-L-NODE effectively operates on high-dimensional spatial-temporal data, we include it as a baseline for a more comprehensive empirical comparison.

Since the original work only considers spatially 2D input, we modify the encoder and decoder from stacked 2D CNNs to 1D CNNs, tune the latent dimension in {4, 8, 16, 64}, and the CNN base output dimension in {16, 64, 128}. Important hyperparameters are reported in Table 31 and the rest are kept the same as in Iakovlev et al. (2023b).

## K 2D SIMULATION BASELINES

### K.1 FNO

In 2D experiments, we train the FNO model using the density, velocity, control, and percentage of smoke through the target bucket of the previous step as the input and using the rest step's density, velocity, and percentage of smoke through the target bucket as the output. The parameters are outlined in Table 32.

Table 31: **Hyperparameters of MS-L-NODE.**

| Hyperparameter name | Value |
|---|---|
| Encoder | |
| Encoder CNN channels | 128 |
| Latent dimension | 8 |
| Decoder | |
| Encoder CNN channels | 128 |
| Latent dimension | 8 |
| Aggregation Network | |
| Heads | 16 |
| Static layers | 4 |
| Dynamical layers | 8 |
| Training | |
| Training batch size | 64 |
| Training iterations | 37500 |
| Learning rate | 1e-3 |

Table 32: **Hyperparameters of 2D FNO.**

| Hyperparameter name | Value |
|---|---|
| Hyperparameters for 2D incompressible fluid: | |
| Number of modes to keep in Fourier Layer | 16 |
| Width of the FNO (*i.e.* number of channels) | 64 |
| Number of input channel | 6 |
| Number of output channel | 4 |
| Number of hidden channels of the lifting block | 256 |
| Number of hidden channels of the projection block | 256 |
| Number of Fourier Layers | 4 |
| Expansion parameter of MLP layer | 0.5 |
| Non-Linearity module | Gelu |
| Rank of the tensor factorization of the Fourier weights | 1.0 |
| Mode of domain padding | one-sided |
| Learning rate | $1 \times 10^{-4}$ |
| Optimizer | Adam |
| Training epochs | 1000 |
| Learning rate scheduler | Cosine |
| Training batch size | 50 |

Table 33: **Hyperparameters of 2D WNO.**

| Hyperparameter name | Value |
|---|---|
| Hyperparameters of the model architecture | |
| Type of wavelet | db4 |
| Level of wavelet decomposition | 2 |
| Uplifting dimension | 8 |
| Number of wavelet layers | 3 |
| Training | |
| Training batch size | 50 |
| Optimizer | Adam |
| Learning rate | 0.05 |
| Training epochs | 500 |
| Learning rate scheduler | StepLR |

## K.2 WNO

The hyperparameters of WNO for the 2D simulation task are in Table 33. And the wavelet we choose is 'bior1.3'.

## K.3 MWT

For a more comprehensive comparison with other wavelet-based approaches, we focus primarily on implementing the 2D baseline inspired by the work in Gupta et al. (2021). Following the previous work, we select the 'Legendre' wavelet. More details on the configurations can be found in Table 34.

Table 34: **Configuration of 2D MWT.**

| Hyperparameter name | Value |
|---|---|
| Wavelet basis | legendre |
| Number of Fourier modes | 12 |
| Kernel size | 3 |
| Training batch size | 200 |
| Training epochs | 300 |
| Optimizer | Adam |
| Learning rate scheduler | MultiStepLR |

Table 35: **Hyperparameters of 2D OFormer.**

| Hyperparameter name | Value |
|---|---|
| Encoder | |
| Type | SpatialTemporalEncoder2D |
| Input Channels | 34 |
| Embedding Dim of Token | 96 |
| Embedding Dim of Encoded Sequence | 192 |
| Heads | 1 |
| Depth | 5 |
| Decoder | |
| Type | PointWiseDecoder2D |
| Out Channels | 1 |
| Embedding Dim of Token | 96 |
| Propagate forward | 1 |
| Length of output sequence | 32 |
| Propagator depth | 1 |
| Curriculum ratio | 0.16 |
| Curriculum steps | 10 |
| Training | |
| Training batch size | 8 |
| Iteration | 100000 |
| Learning rate | 1e-4 |

## K.4 OFORMER

The Operator Transformer (OFormer) Li et al. (2023) is an attention-based framework for learning solution operators of partial differential equations using self-attention, cross-attention, and point-wise MLPs, designed to handle various input sampling patterns and query locations. More details on the configurations can be found in Table 35.

## L    BROADER IMPACTS

Our research proposes a method to simulate and control complex physical systems. We believe our research will bring in significant progress for various scientific and engineering domains, including climate forecasting, fluid control, robotic control, et al. However, there is also a potential that the method might be abused to incur negative social consequences, upon which we should remain vigilant.

