# OpenReview forum: "Wavelet Diffusion Neural Operator"
_ICLR.cc/2025/Conference — ICLR 2025 Poster_

### Official Review · Reviewer_rcJb · 2024-10-27

**Soundness:** 3
**Presentation:** 2
**Contribution:** 3
**Rating:** 8
**Confidence:** 3

**Summary:**

The authors propose the Wavelet Diffusion Neural Operator , a data-driven framework designed for PDE simulation and control. WDNO uses diffusion-based generative modeling within the wavelet domain, which is claimed to capture entire trajectories of the system dynamics. To address the  challenge of generalizing across different resolutions multi-resolution training approach is introduced, enabling WDNO to work across varying data scales. This framework represents a tool for precise, adaptable modeling in complex, multi-scale systems.

**Strengths:**

The paper presents an interesting and rich contribution to the field and offers a new approach through the Wavelet Diffusion Neural Operator. This work is both technically rigorous and well-articulated, making the methodology and results accessible to the reader. The introduction of wavelets is notable, as it appears to deliver the claimed advantages in handling multi-scale features, abrupt changes, and long-term dependencies in PDE-based simulations. These claims are convincingly supported by a number of experiments, which effectively demonstrate the impact of the wavelet-based approach.

**Weaknesses:**

The structure of the work can be challenging to follow, partially due to the number of innovations introduced in quick succession, which may benefit from clearer organization or further segmentation for clarity.

The experimental evaluation, while well-executed and featuring comparisons to other methods, is somewhat limited in the number of test cases. Expanding the range of test cases would enhance the generalizability of the results:
1. An analysis of the error distribution over time would be valuable, as it could reveal any systematic errors or trends in performance decay over extended simulations.
2. Lack of a real-world test case. Please include a real-world test case to provide further validation of the approach in practical settings:  a dataset like PTB-XL should be addressable with the proposed method and could offer relevant, real-world complexity.
3. Additional 2D cases across a broader range of complex spatial-temporal domains would strengthen the  work. [Iakolev et al, Latent neural ODEs with sparse bayesian multiple shooting, ICLR 2023, Lagemann et al, Invariance-based Learning of Latent Dynamics, ICLR 2024] are presenting results for 2D PDE-based test cases. It makes also sense to compare WDNO against these neural ODE based approaches for the simulation part.

Figures 1 and 2 could be improved for clarity and relevance.
In my opinion Figure 1 lacks meaningful content and could be more effectively repurposed. The caption offers minimal context, leaving the figure’s purpose unclear, especially as the main text already conveys WDNO’s super-resolution capabilities. Reallocating this space to highlight other aspects of the method might provide more value.
Figure 2 is informative and well-designed but would benefit from an expanded caption to guide readers through its details, despite space constraints. I would suggest a more general, high-level version of this figure in the main text and the current version of Figure 2 with a detailed caption in the Appendix. This would offer both an accessible overview and a richer, in-depth explanation in the Appendix.
A detailed explanation tracing the steps from input data to output data of the pipeline and the transformations involved in each stage, with explicit dimensions based on one of the datasets would be great and improve the clarity.

In my opinion, eq. 9 and 10 appear not important in the main discussion and could be more appropriately relocated to the corresponding Appendix (E, F, G). Instead, a general paragraph on data preparation in the main text would provide readers with a clearer understanding of the processes involved, which is currently lacking.

Additionally, the work presented in Appendix C is noteworthy and should at least be mentioned briefly in the main text.

Please reorganize the Appendix to ensure that tables are placed immediately after they are referenced. As there is no page limit for the Appendix, there is no need to conserve space, which would increase the readability.

Please add an arrow indicating the direction of time on the time axis in Figure 4.

**Questions:**

Line 194, “ we combine the use of both guidance methods” Where is the classifier based guidance method used?

What is the impact of measurement noise? An ablation study for datasets with increasing measurement noise would be great.

How many trajectories are required? Can you show the impact of the available number of trajectories on the performance?

What happens if the dynamical changes due to interventions/perturbation upon the system? Is the learned model still useful?

---

> ### Author Response · Authors · 2024-11-21
> **Official Response to Reviewer rcJb (1)**
>
> Thank you for recognizing our work as interesting, technically rigorous, well-articulated, and convincing. Below are our responses to your comments.
>
> >Q1: The structure of the work can be challenging to follow, partially due to the number of innovations introduced in quick succession, which may benefit from clearer organization or further segmentation for clarity.
> - Thanks for the suggestion. To make the structure of the paper clearer and easier to follow, at the beginning of the Method section in the revised manuscript, we **add a specific introduction to each subsection**. Additionally, we further organize the content in Section 3.1 and 3.2 of Method by **adding more detailed subheadings**.
>
> >Q2: The experimental evaluation, while well-executed and featuring comparisons to other methods, is somewhat limited in the number of test cases. Expanding the range of test cases would enhance the generalizability of the results:
> >1. An analysis of the error distribution over time would be valuable, as it could reveal any systematic errors or trends in performance decay over extended simulations.
> >2. Lack of a real-world test case. Please include a real-world test case to provide further validation of the approach in practical settings: a dataset like PTB-XL should be addressable with the proposed method and could offer relevant, real-world complexity.
> >3. Additional 2D cases across a broader range of complex spatial-temporal domains would strengthen the work. [Iakolev et al, Latent neural ODEs with sparse bayesian multiple shooting, ICLR 2023, Lagemann et al, Invariance-based Learning of Latent Dynamics, ICLR 2024] are presenting results for 2D PDE-based test cases. It makes also sense to compare WDNO against these neural ODE based approaches for the simulation part.
> - In the ablation study (Figure 5a) in the original submission, we **have already provided the errors of U-Net and WDNO at different timesteps**, pointing out that the MSE of U-Net increases much faster than that of WDNO. To further support this, we **add the errors of other methods at different timesteps**. The results in the table below show that WDNO exhibits the slowest error growth, confirming its ability to capture **long-term dependencies**. We have updated Figure 5a and analysis in Section 4.7 of the updated manuscript accordingly.
> | Time step | 1 | 11 | 21 | 31 |
> | --- | --- | --- | --- | --- |
> | FNO | 0.0019 | 0.0094 | 0.0076 | 0.0034 |
> | OFormer | 0.0113 | 0.0226 | 0.0311 | 0.1138 |
> | MWT | 0.0110 | 0.0242 | 0.0209 | 0.0706 |
> | WNO | 0.1060 | 0.1252 | 0.1158 | 0.0826 |
> | U-Net | 0.0038 | 0.0053 | 0.0099 | 0.0306 |
> | DDPM | 0.0018 | 0.0034 | 0.0133 | 0.0173 |
> | WDNO | 0.0011 | 0.0012 | 0.0031 | 0.0049 |
> - To better demonstrate the capabilities of WDNO, we add experiments on the **challenging real-world weather prediction dataset ERA5** [1], which has a **higher dimensionality** compared to PTB-XL. Compared to the standard task of predicting the next 12 hours based on 12 hours of initial conditions [2], we increase the difficulty by **extending the prediction horizon** to 20 hours. The experimental results in the table below show that WDNO still achieves the best performance. Notably, its relative $L_2$ error for the 20-hour prediction is as low as 0.0161, highlighting its superior capabilities. Here we experiment with different parameters for WNO, but all configurations fail to converge. In the updated manuscript, we incorporate these results into Section 4.5 and Table 1.
> |  | MSE |
> | --- | --- |
> | WNO | - |
> | MWT | 21.85750 |
> | OFormer | 18.26230 |
> | FNO | 14.38638 |
> | U-Net | 15.51342 |
> | DDPM | 15.21103 |
> | WDNO | **12.83291** |
> - First, in the revised manuscript, we **have cited these two references to the first paragraph of Related Work**. Second, we **compare with MSVI [3]** to compare WDNO with neural ODE based approaches. The results of WDNO on 1D compressible Navier-Stokes equation are presented below. It is obvious that WDNO has superior performance. The results have been added to Table 5 in the updated manuscript.
> |  | MSE | MAE | $L_{\infty}$ error |
> | --- | --- | --- | --- |
> | MSVI | 1.7063 | 0.6047 | 17.0386 |
> | WDNO | **0.2195** | **0.1049** | **13.0626** |
>
> [1] Kalnay E, Kanamitsu M, Kistler R, et al. The NCEP/NCAR 40-year reanalysis project[M]. Renewable energy. Routledge, 2018: Vol1_146-Vol1_194.
>
> [2] Tan C, Li S, Gao Z, et al. Openstl: A comprehensive benchmark of spatio-temporal predictive learning[J]. Advances in Neural Information Processing Systems, 2023, 36: 69819-69831.
>
> [3] Iakovlev V, Yildiz C, Heinonen M, et al. Latent Neural ODEs with Sparse Bayesian Multiple Shooting[C]. The Eleventh International Conference on Learning Representations.

---

> > ### Author Response · Authors · 2024-11-21
> > **Official Response to Reviewer rcJb (2)**
> >
> > >Q3: Figures 1 and 2 could be improved for clarity and relevance. In my opinion Figure 1 lacks meaningful content and could be more effectively repurposed. The caption offers minimal context, leaving the figure's purpose unclear, especially as the main text already conveys WDNO's super-resolution capabilities. Reallocating this space to highlight other aspects of the method might provide more value. Figure 2 is informative and well-designed but would benefit from an expanded caption to guide readers through its details, despite space constraints. I would suggest a more general, high-level version of this figure in the main text and the current version of Figure 2 with a detailed caption in the Appendix. This would offer both an accessible overview and a richer, in-depth explanation in the Appendix. A detailed explanation tracing the steps from input data to output data of the pipeline and the transformations involved in each stage, with explicit dimensions based on one of the datasets would be great and improve the clarity.
> > - Thank you for your thoughtful suggestions. Based on your feedback, we have redrawn two figures: **a more general, high-level version** of the original Figure 2 and **a detailed version** of the original Figure 2. The former corresponds to Figure 1 in the updated manuscript, while the latter is included in the appendix as Figure 10.
> > - In the new Figure 1, we create a more **concise and unified figure for BRM and SRM**, which more intuitively illustrates our idea. We **reduce specific details**, such as the exact shape of the data, to make the figure more concise. Additionally, instead of using blocks to represent data as before, we now **utilize real weather data from ERA5 and its actual wavelet transform results**.
> > - In Figure 10, following your suggstion, the figure has been revised to align with the training and inference process of the **1D Burgers' equation**. We plot the **explicit dimensions** based its datasets, **expand the textual explanations** in the figure and add **purple arrows** to clearly distinguish the model's condition.
> >
> > >Q4: In my opinion, Eq. 9 and 10 appear not important in the main discussion and could be more appropriately relocated to the corresponding Appendix (E, F, G). Instead, a general paragraph on data preparation in the main text would provide readers with a clearer understanding of the processes involved, which is currently lacking.
> > - Thanks for the suggestion to help improve readability. We move Eq. 7 and 9 and 10 to the appendix. Additionally, we add descriptions of data preparation to each subsection of the main experiments in Experiment
> >
> > >Q5: The work presented in Appendix C is noteworthy and should at least be mentioned briefly in the main text.
> > - Thank you for the suggestion. We summarize the experimental content from Appendix C in the last paragraph of Ablation Study in the updated manuscript.
> >
> > >Q6: Please reorganize the Appendix to ensure that tables are placed immediately after they are referenced. As there is no page limit for the Appendix, there is no need to conserve space, which would increase the readability.
> > - Thank you for your feedback. We have revised the layout of the tables in the appendix to ensure that each table appears immediately after the paragraph in which it is referenced.
> >
> > >Q7: Please add an arrow indicating the direction of time on the time axis in Figure 4.
> > - Thank you for the suggestion. We have added an arrow in Figure 3 (Figure 4 in the original submission) to indicate the direction of time.
> >
> > >Q8: Line 194, " we combine the use of both guidance methods" Where is the classifier based guidance method used?
> > - As referenced in Section 3.1 in the original submission, "WDNO for Control", we use a classifier-based guidance method for control problems. Specifically, the control objective $\mathcal{J}$ acts as the classifier, and through the classifier-based guidance method, we guide the model to generate control sequences that achieve lower values of $\mathcal{J}$.
> >
> > >Q9: How many trajectories are required? Can you show the impact of the available number of trajectories on the performance?
> > - We conduct experiments on the 1D compressible Navier-Stokes equation by **reducing the training dataset size to 0.2, 0.4, 0.6, and 0.8 times** the current size (9000 samples) and measure WDNO's MSE. The results show that even when the dataset size is reduced to 0.4 times, WDNO's error remains within a relatively small range. When the dataset size is reduced to 0.2 times, the error shows a noticeable increase. In the updated manuscript, we have added results in Ablation Study.
> > | Ratio of data for training | MSE |
> > | --- | --- |
> > | 0.2 | 3.8832 |
> > | 0.4 | 0.6614 |
> > | 0.6 | 0.3239 |
> > | 0.8 | 0.2617 |
> > | 1 | 0.2195 |

---

> > > ### Author Response · Authors · 2024-11-21
> > > **Official Response to Reviewer rcJb (3)**
> > >
> > > >Q10: What is the impact of measurement noise? An ablation study for datasets with increasing measurement noise would be great.
> > > - In **Table 8 of Appendix C.4 in the original submission**, we have included results for **introducing random noise into the control sequence**. The results demonstrate that WDNO still outperforms all other methods even in the presence of noise.
> > > - For scenarios where **noise exists in the observed system state**, to both the training and testing datasets of 1D Burgers' equation, we have added noise, sampled as **Gaussian noise scaled by the original data's standard deviation multiplied by a scale factor**. We test scale factors of 0.01, 0.001, and 0.0001. As shown in the table below, WDNO's results exhibit minimal variation with changes in scale, demonstrating its **robustness to noise**. These results have been incorporated in Ablation Study in the updated manuscript.
> > > |  | 0.01 | 0.001 | 0.0001 | 0 |
> > > | --- | --- | --- | --- | --- |
> > > | WDNO | 0.00021 | 0.00017 | 0.00015 | 0.00014 |
> > >
> > > >Q11: What happens if the dynamical changes due to interventions/perturbation upon the system? Is the learned model still useful?
> > > - As our setting is taking the initial condition and parameters (such as force terms) as input to predict the entire trajectory in a single inference, **the setting we consider does not include real-time feedback from the environment**. Therefore, there is no additional input to inform the model of changes in the system, which then influences the prediction results of WDNO and all the baselines.
> > > - However, to address dynamical changes, we can **incorporate feedback from the external environment** into the models' inference, and we point out that **incorporating feedback from the environment and our proposed method** are **independent directions**. Many studies focus specifically on introducing environmental feedback into the modeling process of diffusion models, and **these approaches can also be integrated with our proposed method**. For example, some methods propose incorporating external feedback through effective replanning with diffusion models [1], while others achieve this by introducing a asynchronous denoising framework [2].
> > >
> > > [1] Zhou S, Du Y, Zhang S, et al. Adaptive online replanning with diffusion models[J]. Advances in Neural Information Processing Systems, 2024, 36.
> > >
> > > [2] Wei L, Feng H, Yang Y, et al. Closed-loop diffusion control of complex physical systems[J]. arXiv preprint arXiv:2408.03124, 2024.
> > >
> > > Once again, thank you for your helpful and insightful suggestions. We hope we have addressed all the issues you raised. Please feel free to reach out to us if there is anything else you wish to discuss.

---

> > > > ### Author Response · Authors · 2024-11-25
> > > > **A gentle reminder: please respond to our rebuttal**
> > > >
> > > > Dear Reviewer rcJb,
> > > >
> > > > Thank you for your time and effort in reviewing our work. We have carefully considered your detailed comments and questions, and we have tried to address all your concerns accordingly.
> > > >
> > > > As the deadline for author-reviewer discussions is approaching, could you please go over our responses? If you find our responses satisfactory, we hope you could consider adjusting your initial rating. Please feel free to share any additional comments you may have.
> > > >
> > > > Thank you!
> > > >
> > > > Authors

---

> > > > > ### Comment · Reviewer_rcJb · 2024-11-25
> > > > >
> > > > > Thank you very much for clarifying my concerns. The rebuttal substantially improved the paper and I raise my score.

---

> > > > > > ### Author Response · Authors · 2024-11-26
> > > > > > **Official Response to Reviewer rcJb**
> > > > > >
> > > > > > Thank you for your instructive suggestions. We are delighted that your feedback has significantly improved our paper!

---

### Official Review · Reviewer_NMDB · 2024-11-01

**Soundness:** 3
**Presentation:** 3
**Contribution:** 2
**Rating:** 6
**Confidence:** 4

**Summary:**

The standard diffusion models cannot learn dynamics with abrupt changes. This paper proposes a new method, Wavelet Diffusion Neural Operator (WDNO), to solve this issue. It combines wavelet transform and diffusion models in the context of learning PDEs from the perspective of neural operators. This method also considers multi-resolution training. Multiple datasets have been tested to evaluate the model performance compared to the baseline models.

**Strengths:**

- This paper aims to solve an interesting problem regarding the abrupt changes in spatiotemporal dynamics.

- This paper has shown comprehensive details of the experimental setup, model details, and results.

- This paper is easy to follow.

**Weaknesses:**

- This paper shows incremental novelty. It uses diffusion models in the wavelet space. But that is not a big issue as long as the model performance is good.

- The motivation for using diffusion models for PDE learning is not well established. This paper mentions that diffusion models can better capture long-term predictions and model high-dimensional systems. However, many deep learning-based models can do both, such as Fourier neural operators. How do diffusion models differentiate from other deep learning-based models? Also, can you provide a reference paper that shows diffusion models can do a better job of capturing long-term predictions?

- I have some concerns about the datasets used in this paper. 1D Burgers with a viscosity of 0.01 is not particularly challenging, though it presents shock wave phenomena. The standard physics-informed neural networks [1] are also capable of handling such tasks. The authors may consider some other challenging datasets, such as ERA5 [2], since the proposed method can deal with long-term predictions. Second, it would also be interesting to see the model performance on the dynamics without abrupt changes. I think that will also give the audience a broader sense of how this model works. The authors may consider some datasets in PDEBench [3].

- It would be good to have another ablation study on the DDPM + Fourier transform, which changes the wavelet transform in the proposed method. The additional experiment will be trained in Fourier space instead of wavelet space. It will further validate the effectiveness of using wavelet transform to capture local details.

- The presentation for Section 2 Related Work can be improved. It would be better to have several paragraphs to introduce the related works from different perspectives.

**References:**

[1] Raissi, M., Perdikaris, P., & Karniadakis, G. E. (2019). Physics-informed neural networks: A deep learning framework for solving forward and inverse problems involving nonlinear partial differential equations. Journal of Computational Physics, 378, 686-707.

[2] Pathak, J., Subramanian, S., Harrington, P., Raja, S., Chattopadhyay, A., Mardani, M., ... & Anandkumar, A. (2022). Fourcastnet: A global data-driven high-resolution weather model using adaptive fourier neural operators. arXiv preprint arXiv:2202.11214.

[3] Takamoto, M., Praditia, T., Leiteritz, R., MacKinlay, D., Alesiani, F., Pflüger, D., & Niepert, M. (2022). Pdebench: An extensive benchmark for scientific machine learning. Advances in Neural Information Processing Systems, 35, 1596-1611.

**Questions:**

- For the evaluation metrics, why does this paper only consider MSE? If this paper focuses on the dynamics with abrupt changes, then the Mean Absolute Error (MAE) or infinity norm should be considered.

- Some minor typos:

    - On Page 2, the paragraph name “wavelet domain” should be “Wavelet domain”.

    - On Page 2,  it seems not common to see “contribute the following”.

---

> ### Author Response · Authors · 2024-11-21
> **Official Response to Reviewer NMDB (1)**
>
> Thanks for the detailed comments. Here are responses to your concerns.
>
> >Q1: This paper shows incremental novelty. It uses diffusion models in the wavelet space. But that is not a big issue as long as the model performance is good.
> - **Achieving "good model performance" is inherently challenging**, which is why special design is necessary. Learning complex dynamics, such as systems with abrupt changes, is particularly difficult. This is evident in two aspects:
>   - First, **existing literature** indicates that deep neural networks tend to prioritize low-frequency signals, leading to poor learning of high-frequency signals like abrupt changes [1].
>   - Second, as shown in **our experiments** through tables and many visualizations in the original submission, other models, including DDPM, struggle to achieve "good model performance." **Figures 2a, 6, 7, and 9** clearly illustrate how WDNO outperforms DDPM in capturing signals with abrupt changes.
> - Besides, to address the fundamental requirement for neural PDE solvers to generalize across different resolutions, we also introduce the **multi-resolution training** seamlessly integrated with wavelets, demonstrating superior performance compared to previous super-resolution baselines in experiments. Additionally, we **unify simulation and control tasks** within one single framework, **highlighting the utility of diffusion models in the PDE domain**.
>
> [1] Xu, Zhi-Qin John, et al. "Frequency principle: Fourier analysis sheds light on deep neural networks." arXiv preprint arXiv:1901.06523.
>
> >Q2: The motivation for using diffusion models for PDE learning is not well established. This paper mentions that diffusion models can better capture long-term predictions and model high-dimensional systems. However, many deep learning-based models can do both, such as Fourier neural operators. How do diffusion models differentiate from other deep learning-based models? Also, can you provide a reference paper that shows diffusion models can do a better job of capturing long-term predictions?
> - Firstly, both in simulation and control, studies **have highlighted** that diffusion models can help capture **long-range dependencies**. In **simulation**, the **noise-learning mechanism** enhances temporal stability, a conclusion widely validated in fluid prediction [1], long-term human motion generation [2] and weather forecasting [3]. Some works even propose, inspired by DDPM, introducing an adapted Gaussian denoising step for stable rollouts in PDE simulation [4]. In **control**, many RL-related studies [5, 6, 7] suggest using diffusion models to model entire trajectories from a **global perspective**, enabling trajectory-level optimization. Thus, this approach facilitates non-greedy planning and achieves near-globally optimal plans more effectively.
> - Secondly, diffusion models are **widely recognized** for their strong modeling capabilities in **complex and high-dimensional systems**, including 3D fluids [8], weather [3], videos [9], 3D shape generation [10] and real-world robots control [5], among others. This advantage stems from its **denoising and noising mechanism**, which not only transforms modeling the original data into predicting simpler noise, but also decomposes the task of modeling a complex system into multiple simpler subtasks through multi-step denoising.
> - Thanks for the suggestion. In the updated manuscript, we have included the references in the second paragraph of Introduction and the last paragraph of Related Work.
>
> [1] Kohl G, et al. Benchmarking autoregressive conditional diffusion models for turbulent flow simulation[C]. ICML AI for Science Workshop, 2024.
>
> [2] Yang Z, et al. Synthesizing long-term human motions with diffusion models via coherent sampling[C]. Proceedings of the 31st ACM International Conference on Multimedia, 2023.
>
> [3] Rühling Cachay S, et al. Dyffusion: A dynamics-informed diffusion model for spatiotemporal forecasting[J]. Advances in Neural Information Processing Systems, 2024, 36.
>
> [4] Lippe P, et al. Pde-refiner: Achieving accurate long rollouts with neural pde solvers[J]. Advances in Neural Information Processing Systems, 2024, 36.
>
> [5] Janner M, Tenenbaum J, et al. Planning with Diffusion for Flexible Behavior Synthesis[C]. International Conference on Machine Learning. PMLR, 2022.
>
> [6] Chi C, et al. Diffusion policy: Visuomotor policy learning via action diffusion[J]. The International Journal of Robotics Research, 2023.
>
> [7] Wei, Long, et al. A Generative Approach to Control Complex Physical Systems.[C] Advances in Neural Information Processing Systems, 2024.
>
> [8] Li T, Biferale L, et al. Synthetic Lagrangian turbulence by generative diffusion models[J]. Nature Machine Intelligence, 2024: 1-11.
>
> [9] Ho J, et al. Video diffusion models[J]. Advances in Neural Information Processing Systems, 2022.
>
> [10] Vahdat A, et al. Lion: Latent point diffusion models for 3d shape generation[J]. Advances in Neural Information Processing Systems, 2022.

---

> > ### Author Response · Authors · 2024-11-21
> > **Official Response to Reviewer NMDB (2)**
> >
> > >Q3: I have some concerns about the datasets used in this paper. 1D Burgers with a viscosity of 0.01 is not particularly challenging, though it presents shock wave phenomena. The standard physics-informed neural networks are also capable of handling such tasks. The authors may consider some other challenging datasets, such as ERA5, since the proposed method can deal with long-term predictions. Second, it would also be interesting to see the model performance on the dynamics without abrupt changes. I think that will also give the audience a broader sense of how this model works. The authors may consider some datasets in PDEBench.
> > - We note that the viscosity in the 1D Burgers dataset follows **previous works** [1, 2]. To further increase the difficulty and demonstrate the effectiveness of our method, in the original submission, we **have already extended the timesteps** from 10 in these works to 80, which means we predict 80 future timesteps based on the initial conditions at a single moment.
> > - Additionally, beyond this experiment, we have tested our method on **two challenging datasets**: the dataset corresponding to the **most difficult** parameter set in the 1D CFD dataset from PDEBench and the **most challenging** 2D experiment from [1].
> > - To further verify our method, we add experimental results on the **challenging ERA5 dataset in Section 4.5 in the updated manuscript**. Unlike the typical setup of using 12 hours of input to predict 12 hours of output [3], we evaluate **long-term predictions** by using 12 hours of input to predict 20 hours of output. The results on WDNO and baselines are provided in the table below. Here we experiment with different parameters for WNO, but all fail to converge. WDNO still achieves the best performance, with a relative $L_2$ error as low as 0.0161, demonstrating its outstanding capability on challenging datasets. We have added these in Section 4.5 in the revised manuscript.
> > |  | MSE |
> > | --- | --- |
> > | WNO | - |
> > | MWT | 21.85750 |
> > | OFormer | 18.26230 |
> > | FNO | 14.38638 |
> > | U-Net | 15.51342 |
> > | DDPM | 15.21103 |
> > | WDNO | **12.83291** |
> > - Thanks for the suggestion. To see the **model performance on the dynamics without abrupt changes**, we conduct tests on the **1D advection equation** dataset in **PDEBench**, which features relatively simple and smooth dynamics. The results are shown below. While different models generally perform well, **WDNO still achieves the best results**. The results are included in Section 4.2 and Table 1 in the updated manuscript.
> > | Method | MSE |
> > | --- | --- |
> > | WNO | 4.216e-02 |
> > | MWT | 3.468e-04 |
> > | CNN | 5.033e-04 |
> > | OFormer | 1.858e-04 |
> > | FNO | 9.712e-04 |
> > | DDPM | 4.209e-05 |
> > | WDNO | **2.898e-05** |
> >
> > [1] Long, Wei, Peiyan Hu, Ruiqi Feng, Haodong Feng, Yixuan Du, Tao Zhang, Rui Wang, Yue Wang, Zhi-Ming Ma, and Tailin Wu. A Generative Approach to Control Complex Physical Systems.[C] Advances in Neural Information Processing Systems, 2024, 36.
> >
> > [2] Hwang R, Lee J Y, Shin J Y, et al. Solving pde-constrained control problems using operator learning[C]. Proceedings of the AAAI Conference on Artificial Intelligence. 2022, 36(4): 4504-4512.
> >
> > [3] Tan C, Li S, Gao Z, et al. Openstl: A comprehensive benchmark of spatio-temporal predictive learning[J]. Advances in Neural Information Processing Systems, 2023, 36: 69819-69831.
> >
> > >Q4: It would be good to have another ablation study on the DDPM + Fourier transform, which changes the wavelet transform in the proposed method. The additional experiment will be trained in Fourier space instead of wavelet space. It will further validate the effectiveness of using wavelet transform to capture local details.
> > - Thanks for the suggestion. To further validate the wavelet transform's ability to capture local details, we provide the MSE, MAE, and $L_\infty$ error of **DDPM combined with Fourier transform** on the 1D compressible Navier-Stokes equation. The implementation is identical to WDNO, except that the wavelet transform is replaced by Fourier transform, implemented using PyTorch's Fast Fourier Transform function. The results show that while combining DDPM with Fourier transform **improves over the original DDPM**, its performance **falls far short of WDNO**. This strongly supports the conclusion that wavelet transforms are indeed beneficial for modeling dynamics with abrupt changes. We have incorporated these results into Section 4.7, Table 5, and Figure 5c in the updated manuscript.
> > |  | MSE | MAE | $L_\infty$ error |
> > | --- | --- | --- | --- |
> > | DDPM | 5.5228 | 0.9795 | 16.0532 |
> > | In the Fourier domain | 3.0258 | 0.8498 | 14.6670 |
> > | WDNO | **0.2195** | **0.1049** | **13.0626** |

---

> > > ### Author Response · Authors · 2024-11-21
> > > **Official Response to Reviewer NMDB (3)**
> > >
> > > >Q5: The presentation for Section 2 Related Work can be improved. It would be better to have several paragraphs to introduce the related works from different perspectives.
> > > - The previous version of the Related Work section contained three paragraphs discussing PDE simulation, PDE control, and diffusion models. Based on your suggestion, we have **revised it into 6 paragraphs** and added **subheadings at the beginning of each paragraph** for clarity. The six paragraphs now consist of the original three sections, along with the newly added ones: Super-resolution tasks, Wavelet transform, and Long-term predictions.
> > >
> > > >Q6: For the evaluation metrics, why does this paper only consider MSE? If this paper focuses on the dynamics with abrupt changes, then the Mean Absolute Error (MAE) or infinite norm should be considered.
> > > - Previously, we demonstrated the ability to model abrupt changes through **extensive visualizations**, including Figures 2a, 6, 7, and 9. We now additionally provide the **MAE and $L_\infty$ error** of all models on the 1D compressible Navier-Stokes dataset. It can be observed that the trends of MAE align closely with MSE. However, the $L_\infty$ error values across different methods are relatively similar because this metric only considers the maximum value across the entire spatiotemporal domain, thus capturing less information. These have been reported in Table 5 in the updated manuscript.
> > > |  | MSE | MAE | $L_\infty$ error |
> > > | --- | --- | --- | --- |
> > > | WNO | 6.5428 | 1.1921 | 21.3860 |
> > > | MWT | 1.3830 | 0.5196 | 11.3677 |
> > > | OFormer | 0.6227 | 0.4006 | 30.9019 |
> > > | FNO | 0.2575 | 0.1985 | **11.1495** |
> > > | CNN | 12.4966 | 1.2111 | 17.6116 |
> > > | DDPM | 5.5228 | 0.9795 | 16.0532 |
> > > | WDNO | **0.2195** | **0.1049** | 13.0626 |
> > >
> > > >Q7: Some minor typos:
> > > >  - On Page 2, the paragraph name "wavelet domain" should be "Wavelet domain".
> > > >  - On Page 2, it seems not common to see "contribute the following".
> > > - On page 2, we only find "wavelet domain" in bold in the penultimate paragraph of the Introduction. Is this the issue you are referring to? However, it is not a paragraph's name. The complete paragraph name is "Generation in the wavelet domain."
> > > - We have revised "we contribute the following" to "our contributions include the following".
> > >
> > > We have addressed each of your comments. If you have any further concerns, please feel free to reach out to us, and we will be happy to provide additional clarification.

---

> > > > ### Comment · Reviewer_NMDB · 2024-11-24
> > > >
> > > > Thanks for your rebuttal. My concerns have been addressed. I am raising my score.

---

> > > > > ### Author Response · Authors · 2024-11-25
> > > > >
> > > > > Thanks for your valuable feedback and raising the score. We are glad to have addressed your concerns!

---

### Official Review · Reviewer_iVpN · 2024-11-01

**Soundness:** 2
**Presentation:** 2
**Contribution:** 2
**Rating:** 6
**Confidence:** 5

**Summary:**

This paper proposes a wavelet diffusion neural operator (WDNO), which performs diffusion-based generative modeling in the wavelet domain for the entire trajectory of time-dependent PDEs and multiresolution training to generalize across different resolutions.

**Strengths:**

1) Multi-resolution training

2) Diffusion in the wavelet domain to capture long-term dependencies and abrupt changes effectively.

**Weaknesses:**

1) The evaluation omits comparisons with state-of-the-art operators like Transolver, GNOT, LSM, DPOT, and CNO etc

2) Training time, inference speed, and memory usage are not compared for WDNO and baselines.

3) Lacks novelty as compared to previous works such as:

* Benchmarking Autoregressive Conditional Diffusion Models for Turbulent Flow Simulation (arXiv:2309.01745)

* DiffusionPDE: Generative PDE-Solving Under Partial Observation (arXiv:2406.17763)

**Questions:**

1)  How does WDNO compare to baselines regarding training time, memory usage, and inference speed?

2) How was the basis wavelet chosen for the benchmarks?

3) Have the authors investigated using diffusion in the Fourier domain for comparison with WDNO, considering the computational efficiency of Fourier transforms?

4) How were hyperparameters for baselines chosen? Were they optimized fairly compared to WDNO?

5) The paper needs to elaborate on why WDNO outperforms DDPM, considering Parseval's identity, which states that information content remains constant during transformations.

6)  Why wasn't the Wavelet Neural Operator (WNO) included as a baseline for super-resolution tasks? How does WDNO compare to WNO, DDPM, UNET, etc., for long-range dependencies (Figure 6b)?

7) The paper should cite relevant related work and add as baselines, including:

* Benchmarking Autoregressive Conditional Diffusion Models for Turbulent Flow Simulation (arXiv:2309.01745)

* DiffusionPDE: Generative PDE-Solving Under Partial Observation (arXiv:2406.17763)

---

> ### Author Response · Authors · 2024-11-21
> **Official Response to Reviewer iVpN (1)**
>
> We thank the reviewer for the comments and questions. Below we address the reviewer's concerns.
>
> >Q1: The evaluation omits comparisons with state-of-the-art operators like Transolver, GNOT, LSM, DPOT, and CNO etc.
> - Thanks for your suggestions. We previously selected baselines that are **diverse, high-performing, and highly relevant**. These include methods based on wavelet transforms (WNO, MWT), Fourier transforms (FNO), Transformer architectures (OFormer), convolutional networks (CNN, U-Net), and diffusion models (DDPM).
> - Based on your valuable comments, first, we have **cited these methods** in Related Work in the revised manuscript. Second, we have **added the mentioned baselines of Transolver and CNO** to further evaluate the effectiveness of our method. These are chosen because Transolver has been shown to outperform **GNOT and LSM** across various experiments in its paper [1], while **DPOT**, a pre-trained foundation model, requires 10 timesteps to predict the next frame, which is not applicable to our setting where 1 timestep is used to predict the entire trajectory. In experiments on the 1D compressible Navier-Stokes equation, WDNO continues to show the best performance. We have incorporated the results in Table 5 in the updated manuscript.
> |  | MSE | MAE | $L_\infty$ error |
> | --- | --- | --- | --- |
> | Transolver | 4.99843 | 0.40253 | **4.87284** |
> | CNO | 0.3987 | 0.2765 | 9.9169 |
> | ACDM | 4.6574 | 0.8946 | 60.9370 |
> | DiffusionPDE | 5.5936 | 0.9792 | 16.0515 |
> | WNO | 6.5428 | 1.1921 | 21.3860 |
> | MWT | 1.3830 | 0.5196 | 11.3677 |
> | OFormer | 0.6227 | 0.4006 | 30.90186 |
> | FNO | 0.2575 | 0.1985 | 11.1495 |
> | CNN | 12.4966 | 1.2111 | 17.6116 |
> | DDPM | 5.5228 | 0.9795 | 16.0532 |
> | WDNO | **0.2195** | **0.1049** | 13.0626 |
>
> [1] Wu H, Luo H, Wang H, et al. Transolver: A Fast Transformer Solver for PDEs on General Geometries[C]. Forty-first International Conference on Machine Learning.
>
> >Q2:
> > 1. The paper should cite relevant related work and add as baselines, including:
> >    - Benchmarking Autoregressive Conditional Diffusion Models for Turbulent Flow Simulation (arXiv:2309.01745)
> >    - DiffusionPDE: Generative PDE-Solving Under Partial Observation (arXiv:2406.17763)
> > 2. Lacks novelty as compared to these works.
> - Firstly, our work **differs significantly** from these two works. The first work, **ACDM**, uses a diffusion model for autoregressive system state prediction, which fundamentally differs from our approach of modeling the entire trajectory distribution simultaneously. As we have elaborated in our original submission, this global modeling allows the model to better capture long-range dependencies, leading to **higher accuracy in prediction tasks** and enabling **globally optimal planning in control tasks**. Moreover, our work introduces **wavelet transforms** to capture **abrupt changes**, a **multi-resolution framework** for super-resolution, and demonstrates how the entire framework seamlessly applies to control tasks with **superior performance**.
> - As for the second work, **DiffusionPDE** focuses on the scenarios where there lacks full knowledge of the scene. Its algorithmic innovations based on diffusion models are specifically designed for the **partially observed scenario**, which means this work focuses on a different aspect, and its **innovations do not overlap** with ours.
> - Finally, to verify our statement, we **conduct the comparisons** with these two methods. As shown in the above table and the Table 5 in the updated manuscript, the results on 1D compressible Navier-Stokes equation simulation demonstrates that WDNO outperforms them. We also have included them in the discussion of **Related Work**.

---

> ### Author Response · Authors · 2024-11-21
> **Official Response to Reviewer iVpN (2)**
>
> >Q3: Have the authors investigated using diffusion in the Fourier domain for comparison with WDNO, considering the computational efficiency of Fourier transforms?
> - First, we emphasize that **Fourier transforms are not more computationally efficient than wavelet transforms**. We note that the wavelet transform we adopt is parallelizable and can run on GPUs. To demonstrate this, we provide a **speed comparison** in the table below, showing the total time for Fourier and wavelet transforms on the training set (9000 samples) of the 1D compressible Navier-Stokes equation. The Fourier transform is implemented using PyTorch's 2D Fast Fourier Transform function, while the wavelet transform is implemented via PyTorch Wavelets [1] as already mentioned in the Appendix. Both times are recorded on an A100 GPU with a batch size of 2000. It can be observed that the **wavelet transform requires even less time** than the Fourier transform. We have incorporated the results in Table 4 of Appendix A in the updated version.
> | | Wavelet transform | Fourier transform |
> |---|---|---|
> | Time (s) | 1.0171 | 1.3810 |
> - Following your suggestion, we **conduct experiments in the Fourier domain**. The implementation strictly follows WDNO, except for replacing the wavelet transform with Fourier transform. The MSE and MAE results on the 1D compressible Navier-Stokes equation are shown in the table below. We see that while the Fourier transform also provides some improvement over DDPM, its performance is significantly **inferior to that of the wavelet transform**. As already mentioned in the Section 3.1 in the original submission, this is because wavelet transforms inherently decompose information into low-frequency components and high-frequency details across different directions, making them more effective for learning complex system dynamics, such as those with abrupt changes. These results have been incorporated in Figure 5c and Ablation Study of the updated manuscript.
> |  | MSE | MAE |
> | --- | --- | --- |
> | DDPM | 5.5228 | 0.9795 |
> | In the Fourier domain | 3.0258 | 0.8498 |
> | WDNO | **0.2195** | **0.1049** |
>
> [1] Cotter, Fergal. "Uses of Complex Wavelets in Deep Convolutional Neural Networks". Apollo - University of Cambridge Repository, 2019, doi:10.17863/CAM.53748.
>
> >Q4: The paper needs to elaborate on why WDNO outperforms DDPM, considering Parseval's identity, which states that information content remains constant during transformations.
> - Firstly, for different models, the total input information remains constant, yet their performance varies significantly. This discrepancy arises from differences in **model design**, which **impact the effectiveness of learning**.
> - Secondly, existing studies have shown that deep neural networks tend to **prioritize learning low-frequency information** [1]. As a result, they naturally struggle to capture dynamics with abrupt changes, which are common in PDEs, necessitating special designs to address this limitation. To tackle this, as stated in the penultimate paragraph of Section 1 and the first two paragraphs of Section 3.1 in the original submission, **wavelet bases** are localized in both space-time and frequency domains, decomposing the system state into low-frequency and high-frequency information. This naturally **helps models capture the high-frequency components that are typically harder to learn**.
> - Moreover, to analyze why WDNO outperforms DDPM, we **have provided extensive visualizations of the prediction results for DDPM and WDNO** on two equations in **Figures 2a, 6, 7, and 9 in the original submission**. Our analysis clearly demonstrates that WDNO achieves significant improvements in capturing dynamics with abrupt changes, which are otherwise difficult to learn.
>
> [1] Xu, Zhi-Qin John, et al. "Frequency principle: Fourier analysis sheds light on deep neural networks." arXiv preprint arXiv:1901.06523 (2019).

---

> > ### Author Response · Authors · 2024-11-21
> > **Official Response to Reviewer iVpN (3)**
> >
> > >Q5: Why wasn't the Wavelet Neural Operator (WNO) included as a baseline for super-resolution tasks? How does WDNO compare to WNO, DDPM, UNET, etc., for long-range dependencies (Figure 6b)?
> > - We additionally **add WNO's results for the 1D super resolution task** in the table below and update Figure 4 and Section 4.6 in the revised manuscript. It is evident that WNO **performs poorly in super resolution**. As the number of super resolution levels increases, WNO's error grows rapidly, whereas only WDNO achieves error reduction. This demonstrates the superiority of our proposed multi-resolution training.
> > | Level of super resolution | 0 | 1 | 2 | 3 |
> > | --- | --- | --- | --- | --- |
> > | WNO (linear) | 0.0110 | 0.6284 | 1.4474 | 2.0588 |
> > | WDNO (linear) | **0.0026** | **0.0007** | **0.0004** | **0.0004** |
> > | WNO (nearest) | 0.0079 | 0.6491 | 1.5007 | 2.0588 |
> > | WDNO (nearest) | 0.0038 | 0.0011 | 0.0005 | 0.0004 |
> > - As for **2D** super resolution experiments, WNO is **inapplicable** to this specific scenario. It is because WNO achieves super-resolution by adjusting the number of wavelet transform layers, which **simultaneously changes the resolution in both time and space**, but the 2D super resolution experiments focus solely on spatial super resolution.
> > - Thanks for the suggestion. In the original manuscript, we select **U-Net** for comparison because it is the model used in diffusion models for noise prediction. We now add MSE results for other models at different timesteps, from which we can observe that the **WDNO exhibits the slowest error growth** over time. The Ablation Study and Figure 5a is updated in the revised manuscript.
> > | Time step | 1 | 11 | 21 | 31 |
> > | --- | --- | --- | --- | --- |
> > | FNO | 0.0019 | 0.0094 | 0.0076 | 0.0034 |
> > | OFormer | 0.0113 | 0.0226 | 0.0311 | 0.1138 |
> > | MWT | 0.0110 | 0.0242 | 0.0209 | 0.0706 |
> > | WNO | 0.1060 | 0.1252 | 0.1158 | 0.0826 |
> > | U-Net | 0.0038 | 0.0053 | 0.0099 | 0.0306 |
> > | DDPM | 0.0018 | 0.0034 | 0.0133 | 0.0173 |
> > | WDNO | 0.0011 | 0.0012 | 0.0031 | 0.0049 |
> >
> > >Q6: How does WDNO compare to baselines regarding training time, memory usage, and inference speed?
> > - In the original submission, we **have already provided the number of parameters and inference time** of baselines and WDNO of 1D Burgers' equation's in Appendix C.6. Also, in Appendix C.6, we have already provided the **total training time and inference time** of WDNO.
> > - Here, we further provide the **training time** of baselines and WDNO in the table below and Table 13 in the updated manuscript. It can be observed that **WDNO's training time is the least** among all models.
> > |  | Time (h) |
> > | --- | --- |
> > | WNO | 4.5 |
> > | MWT | 6.5 |
> > | OFormer | 19.7 |
> > | FNO | 10.5 |
> > | CNN | 63.8 |
> > | DDPM | 7.8 |
> > | WDNO | 2.5 |
> >
> > >Q7: How was the basis wavelet chosen for the benchmarks?
> > - Thank you for your question. We have added details on the choice of basis wavelet for wavelet-related baselines. We emphasize that we select the wavelet bases that yield the best performance for these baselines to ensure their **optimal results**. These have been added in Appendix J and K in the updated manuscript.
> > |  | WNO | MWT |
> > | --- | --- | --- |
> > | 1D Burgers | sym4 | Legendre |
> > | 1D CFD | bior2.4 | Legendre |
> > | 2D NS | bior1.3 | Legendre |
> >
> > >Q8: How were hyperparameters for baselines chosen? Were they optimized fairly compared to WDNO?
> > - In **Appendix I, J and K in the original submission**, we have already provided the hyperparameters of 1D control, 1D simulation, and 2D simulation baselines, respectively. These hyperparameter correspond to the best results obtained after **searching**, using a **computation budget similar to WDNO**, ensuring a fair comparison. The hyperparameter settings for 2D control baselines follow those from previous work [11].
> >
> > [1] Long, Wei, Peiyan Hu, Ruiqi Feng, Haodong Feng, Yixuan Du, Tao Zhang, Rui Wang, Yue Wang, Zhi-Ming Ma, and Tailin Wu. "A Generative Approach to Control Complex Physical Systems." CoRR (2024).
> >
> > Hope the above have addressed your questions and resolved your concerns. If there is anything else you'd like to discuss, please feel free to reach out, and we will be glad to respond.

---

> > > ### Author Response · Authors · 2024-11-25
> > > **A gentle reminder: please respond to our rebuttal**
> > >
> > > Dear Reviewer iVpN,
> > >
> > > Thank you for your time and effort in reviewing our work. We have carefully considered your detailed comments and questions, and we have tried to address all your concerns accordingly.
> > >
> > > As the deadline for author-reviewer discussions is approaching, could you please go over our responses? If you find our responses satisfactory, we hope you could consider adjusting your initial rating. Please feel free to share any additional comments you may have.
> > >
> > > Thank you!
> > >
> > > Authors

---

> > > > ### Comment · Reviewer_iVpN · 2024-11-25
> > > >
> > > > I want to thank the author for the rebuttal. I still have some queries regarding the response. Please clarify it for me. Also, most of my questions are answered, and I hope to see additional responses in a revised version of the manuscript.
> > > >
> > > > > first work, ACDM, uses a diffusion model for autoregressive system state prediction, which fundamentally differs from our approach of modeling the entire trajectory distribution simultaneously. Can you please elaborate more on the above statement?

---

> ### Author Response · Authors · 2024-11-25
> **Official Response to Reviewer iVpN**
>
> Thank you for your response and constructive discussion. We are glad that we have addressed most of your questions. Below, we provide further clarification on the points you raised.
> - **ACDM** is an autoregressive model, which means both training and inference are performed sequentially.
>   - The model predicts the next $k$ steps $u_{[k,2k-1]}$ based on the previous $k$ steps $u_{[0,k-1]}$. Then, using $u_{[k,2k-1]}$, it predicts $u_{[2k,3k-1]}$, and this process continues iteratively until the entire trajectory $u_{[0,T-1]}$ of $T$ steps is generated. In contrast, our proposed method predicts the entire trajectory $u_{[0,T-1]}$ directly in a single inference step based on the given $k$ initial steps.
>   - For example, in our 1D experiment, where $k=1$ and $T=80$, the model needs to **perform inference 80 times** to predict the entire trajectory, while our proposed method predicts the entire trajectory in **a single inference step**, significantly reducing computational overhead.
> - We note that we **have already made the corresponding modifications to the revised manuscript** in response to your questions, as mentioned in our previous replies. Regarding the **new clarification about ACDM**, we have incorporated this content into the **'Diffusion models' part of Related Work**. We have also uploaded the revised manuscript.
>
> If you have any further points to discuss, please feel free to reach out at any time.

---

> > ### Comment · Reviewer_iVpN · 2024-11-25
> >
> > Thanks for clarifying my query. Going through all the reviewer responses and rebuttals, I have decided to update my score with the hope of seeing all the discussion in the updated version.

---

> > > ### Author Response · Authors · 2024-11-26
> > > **Official Response to Reviewer iVpN**
> > >
> > > Thanks for your valuable feedback on our responses. We genuinely value your contributions and will ensure that any further suggestions will be carefully incorporated.

---

### Official Review · Reviewer_34r3 · 2024-11-02

**Soundness:** 3
**Presentation:** 3
**Contribution:** 3
**Rating:** 8
**Confidence:** 4

**Summary:**

This paper presents a new method to simulate and control physical systems governed by specific PDEs. The method is based on learning a diffusion generative model in the wavelet domain with multi-resolution. The use of wavelet basis is specially convenient for discontinuous behaviours such as shock waves in fluid dynamics, and the diffusion procedure ensures that the full-trajectory error remains bounded. The method is tested with several fluid dynamic problems such as 1D Bruger's equaion, 1D and 2D Navier-Stokes in both simulation and control tasks.

**Strengths:**

* The main idea of combining WNO with diffusion is simple, novel and well justified.
* Every design choice is justified with multiple tests and ablation studies.
* The method outperforms other similar state-of-the-art techniques such as traditional U-Net/CNN to the newest Neural Operator architectures.

**Weaknesses:**

* The method is constrained to static uniform grid data and low-dimensional toy examples.
* The generalization to more complicated tasks or boundary conditions might not be trivial.

**Questions:**

* Can the authors discuss on the implication of using a regular grid? Is this constraint related to the use of a Fast Wavelet Transform? This restriction might be problematic when applying this algorithm to more complex geometries.
* The examples shown on the paper have changing initial conditions, but the geometry and boundaries remain fixed. Can the authors describe how the WDNO architecture is generalizable to other parameters or boundary conditions? It might be interesting for the reader to have a comment over problems with parametric solutions.
* Eq. 4: There is no reference to the guidance rate $\lambda$ until Appendix C3. Can the authors define it next to the equation, to improve readability?

Final comment: The paper has a solid idea, with well justified design choices and the results outperform existing techniques. Based on the comments above, my rating for this paper is marginally above the acceptance threshold.

**Details Of Ethics Concerns:**

I have no ethics concerns.

---

> ### Author Response · Authors · 2024-11-21
> **Official Response to Reviewer 34r3 (1)**
>
> Thank you for complimenting our paper as novel, solid, and high-performing. We have summarized and categorized your questions. Here are our responses to your comments.
>
> >Q1:
> >1. The method is constrained to static uniform grid data.
> >2. Can the authors discuss on the implication of using a regular grid? Is this constraint related to the use of a Fast Wavelet Transform? This restriction might be problematic when applying this algorithm to more complex geometries.
> - A **regular grid** refers to a uniform grid that is structured in a square or rectangular arrangement. **Using a regular grid** means that our architecture cannot learn data with irregular structures like graphs.
> - Currently, the wavelet transform we use does make it challenging to handle data on irregular grids. However, as noted in Section 5 (Limitation and Future Work), our method **could potentially be extended to irregular data**. There are several possible approaches, including using geometric wavelets [1] combined with diffusion models designed for graph structures [2], or projecting data from irregular grids onto regular uniform grids [3, 4], among others.
> - In Section 5 in the updated manuscript, we have added explanation and offered approaches to generalize our method to more complex geometries.
>
> [1] Xu B, Shen H, Cao Q, et al. Graph Wavelet Neural Network[C]. International Conference on Learning Representations. 2018.
>
> [2] Vignac C, Krawczuk I, Siraudin A, et al. DiGress: Discrete Denoising diffusion for graph generation[C]. The Eleventh International Conference on Learning Representations.
>
> [3] Li Z, Huang D Z, Liu B, et al. Fourier neural operator with learned deformations for pdes on general geometries[J]. Journal of Machine Learning Research, 2023, 24(388): 1-26.
>
> [4] Gao H, Sun L, Wang J X. PhyGeoNet: Physics-informed geometry-adaptive convolutional neural networks for solving parameterized steady-state PDEs on irregular domain[J]. Journal of Computational Physics, 2021, 428: 110079.
>
> >Q2:
> >1. The method is constrained to low-dimensional toy examples.
> >2. The generalization to more complicated tasks might not be trivial.
> - To address your concern, we add a **challenging real-world weather prediction dataset**, ERA5 [1], to our experiments. Additionally, we select a more challenging task: **predicting 20 hours ahead based on 12 hours of observations**, instead of the standard 12-hour prediction task [2]. The results in the table show that WDNO still achieves the best performance, with a relative $L_2$ loss as low as 0.0161. And we have experimented with different parameters for WNO, but all configurations fail to converge. These results are included in Section 4.5 and Table 1 in the updated version.
> |  | MSE |
> | --- | --- |
> | WNO | - |
> | MWT | 21.85750 |
> | OFormer | 18.26230 |
> | FNO | 14.38638 |
> | U-Net | 15.51342 |
> | DDPM | 15.21103 |
> | WDNO | **12.83291** |
>
> [1] Kalnay E, Kanamitsu M, Kistler R, et al. The NCEP/NCAR 40-year reanalysis project[M]. Renewable energy. Routledge, 2018: Vol1_146-Vol1_194.
>
> [2] Tan C, Li S, Gao Z, et al. Openstl: A comprehensive benchmark of spatio-temporal predictive learning[J]. Advances in Neural Information Processing Systems, 2023, 36: 69819-69831.

---

> > ### Author Response · Authors · 2024-11-21
> > **Official Response to Reviewer 34r3 (2)**
> >
> > >Q3:
> > >1. The examples shown on the paper have changing initial conditions, but the geometry and boundaries remain fixed. Can the authors describe how the WDNO architecture is generalizable to other parameters or boundary conditions? It might be interesting for the reader to have a comment over problems with parametric solutions.
> > >2. The generalization to more complicated boundary conditions might not be trivial.
> > - **Parameter**: We note that, in addition to the initial conditions, for the 1D Burgers' equation and the 2D Navier-Stokes equation, the **external force term** also varies. This is essentially a special case of parametric solutions and is a more challenging one due to the force term's spatial dimension. For other parameters, we can always transform them into conditions for the diffusion model through techniques such as replication or padding, thereby obtaining parametric solutions.
> > - **Complex boundary conditions**:
> >   - If you are referring to **different types** such as periodic, Dirichlet, or Riemann boundaries, we point out, as mentioned in the previous paragraph, that these are essentially another form of parameters. They can be easily incorporated as conditions of the model. As the boundary condition can be formalized as $\alpha u + \beta\frac{\partial u}{\partial \hat{n}} = f(x)$, we can take $\alpha$, $\beta$ and $f(x)$ (which can be padded to align with the data's shape) as conditions of the diffusion model.
> >   - If this refers to differences in the **geometry** of the boundary, there are also practical solutions. One solution is to use a **mask** to represent the spatial domain as a condition for the diffusion model, assigning a value of 1 inside the domain and 0 outside. Alternatively, following previous work [1], we can use a transformer to map the geometric shape of the boundary into a **latent feature**, which can then serve as a condition of the diffusion model.
> > - We note that the above approaches can be easily integrated with our proposed method.
> >
> > [1] Wang H, Jiaxin LI, Dwivedi A, Hara K, Wu T. BENO: Boundary-embedded Neural Operators for Elliptic PDEs. In The Twelfth International Conference on Learning Representations.
> >
> > >Q4: Eq. 4: There is no reference to the guidance rate λ until Appendix C3. Can the authors define it next to the equation, to improve readability?
> > - Thank you for your suggestion. In the revised manuscript, we have added the definition of $\lambda$ immediately after Eq. 4, where it first appears.
> >
> > Thank you for your instructive suggestions, which have helped us improve the paper. We hope we have addressed your concerns. Please feel free to continue the discussion if anything remains unclear.

---

> > > ### Author Response · Authors · 2024-11-25
> > > **A gentle reminder: please respond to our rebuttal**
> > >
> > > Dear Reviewer 34r3,
> > >
> > > Thank you for your time and effort in reviewing our work. We have carefully considered your detailed comments and questions, and we have tried to address all your concerns accordingly.
> > >
> > > As the deadline for author-reviewer discussions is approaching, could you please go over our responses? If you find our responses satisfactory, we hope you could consider adjusting your initial rating. Please feel free to share any additional comments you may have.
> > >
> > > Thank you!
> > >
> > > Authors

---

> > > > ### Comment · Reviewer_34r3 · 2024-11-25
> > > >
> > > > I would like to thank the authors for the rebuttal: all my concerns have been addressed. I appreciate the effort of including a new example to show that it is applicable to more complex scenarios. I've raised my initial rating.

---

> > > > > ### Author Response · Authors · 2024-11-26
> > > > > **Official Response to Reviewer 34r3**
> > > > >
> > > > > We are glad to hear that all your concerns have been addressed. Thanks also for your valuable feedback and recognition!

---

### Official Review · Reviewer_FYpM · 2024-11-03

**Soundness:** 3
**Presentation:** 2
**Contribution:** 3
**Rating:** 6
**Confidence:** 4

**Summary:**

Aiming at improving the performance of diffusion generative models, in particular for cases with abrupt changes, the authors propose to construct such diffusion models in the wavelet space. To reduce the solution manifold that the model needs to learn and therefore to ease super-resolution, the authors propose to scale the systems such that they align spatio-temporally, and additionally propose to train the model at multiple scales. The resulting model over-performs the state-of-the-art baselines in various simulation and control tasks.

**Strengths:**

1. A wavelet-based neural operator that surpasses the performance of exising methods.
2. Scaling of the input system reduces the solution manifold that the model needs to learn.
3. The paper is generally well-written, with details carefully reported.

**Weaknesses:**

The most significant weakness point of the work is the small number of temporal steps in all the test cases, which might actually be one of the limitation of these models, since the usual approach adopted to achieve long roll-out inference without instability is to introduce artificial training noise - it might be difficult to do so when one is already training a denoising model. It should be noted that the authors iteratively try to strengthen the point that they are doing "long-term dynamics" in the temproal domain, but their reported cases, e.g., the 2D incompressive fluid case, only consist 32 time steps each, which in the commonsense definition is quite a short forecasting window. I strongly suggest removing such terms from the paper to avoid confusion for future readers of the work.

**Questions:**

1. It seems that the Fourier neural operator is second to the proposed method in the simulation tasks. I am curious about how would FNO perform if it is also trained through a similar process as the proposed model (since the original FNO is not trained as a denoising model)?

---

> ### Author Response · Authors · 2024-11-21
> **Official Response to Reviewer FYpM (1)**
>
> Thank you for your recognition of our paper. Below are our responses to your concerns.
>
> >Q1: The most significant weakness point of the work is the small number of temporal steps in all the test cases, which might actually be one of the limitation of these models, since the usual approach adopted to achieve long roll-out inference without instability is to introduce artificial training noise - it might be difficult to do so when one is already training a denoising model. It should be noted that the authors iteratively try to strengthen the point that they are doing "long-term dynamics" in the temproal domain, but their reported cases, e.g., the 2D incompressive fluid case, only consist 32 time steps each, which in the commonsense definition is quite a short forecasting window. I strongly suggest removing such terms from the paper to avoid confusion for future readers of the work.
> - We agree that **adding artificial noise** can help achieve stable long-term predictions for the model. And we believe this is **one of the key reasons why diffusion models are well-suited for long-term predictions**.
>   - This is because the **noise-learning mechanism** in the training process naturally **equips the model with a stronger ability to handle noise**. For instance, as highlighted in [1], inspired by diffusion models, introducing an adapted Gaussian denoising step can enhance the temporal stability of PDE simulations, enabling accurate long roll-outs. Moreover, we point out that the noise addition in diffusion models is **global and more comprehensive** [2], which may lead to even better temporal stability compared to such specially designed approaches.
>   - In addition, **many studies have applied diffusion models in specific domains** and concluded that they are effective for long roll-out inference. These applications span various fields, including fluid prediction [3], weather forecasting [4], decision-making [5], human motion generation[6] and so on.
>   - We have added the mentioned references into Introduction and Related Work in the updated manuscript.
> - Regarding the **timesteps** chosen in our experiments,
>   - We use 80 timesteps for both 1D experiments. Notably, in the 1D Burgers experiment, we **extend the timesteps from 10 to 80** compared to previous studies [7, 8]. For the **super-resolution** task, the total timesteps reach $\mathbf{80 \times 2^3=640}$.
>   - In the 2D experiment, the dataset contains 32 timesteps, but the actual **physical simulation spans 128 timesteps**. Another reason for selecting this timestep is that it corresponds to the point where **the smoke almost disappears from the observation region**.
>   - Our proposed method has the ability to perform longer tasks. However, for 2D problems, the presence of two spatial dimensions limits long-term predictions due to **memory constraints**.
> - In Ablation Study in the original submission, we have already compared the **prediction errors** of WDNO and U-Net **at different timesteps**. To further illustrate this, we add results of other models. As shown in the table below, WDNO exhibits the slowest error growth over time, demonstrating its suitability for long-term predictions. We have updated Figure 5a and the analysis in Ablation Study in the updated manuscript.
> | Time step | 1 | 11 | 21 | 31 |
> | --- | --- | --- | --- | --- |
> | FNO | 0.0019 | 0.0094 | 0.0076 | 0.0034 |
> | OFormer | 0.0113 | 0.0226 | 0.0311 | 0.1138 |
> | MWT | 0.0110 | 0.0242 | 0.0209 | 0.0706 |
> | WNO | 0.1060 | 0.1252 | 0.1158 | 0.0826 |
> | U-Net | 0.0038 | 0.0053 | 0.0099 | 0.0306 |
> | DDPM | 0.0018 | 0.0034 | 0.0133 | 0.0173 |
> | WDNO | 0.0011 | 0.0012 | 0.0031 | 0.0049 |

---

> > ### Author Response · Authors · 2024-11-21
> > **Official Response to Reviewer FYpM (2)**
> >
> > >Q2: It seems that the Fourier neural operator is second to the proposed method in the simulation tasks. I am curious about how would FNO perform if it is also trained through a similar process as the proposed model (since the original FNO is not trained as a denoising model)?
> > - Thank you for your suggestion, which is an interesting idea. We test replacing the noise prediction model from U-Net with FNO. The results in the table below indicate that this is not a good choice, despite adjusting the FNO's number of layers, hidden channels, number of nodes, training steps, denoising steps, and the DDIM parameter $\eta$, and choosing the best one. We believe this may be because FNO tends to filter out high-frequency information, which is crucial for a noise prediction model. In the updated manuscript, we have incorporated the results in Figure 5c and Table 5.
> > |  | MSE | MAE | $L_{\infty}$ error |
> > | --- | --- | --- | --- |
> > | FNO Denoiser | 148.4732 | 6.7660 | 31.9618 |
> > | WDNO | **0.2195** | **0.1049** | **13.0626** |
> > - However, we point out that **the noise prediction model is decoupled from our proposed method**. It is not limited to U-Net or FNO and can be replaced with any model with strong predictive capabilities.
> >
> > The above are our responses to your questions. If there is anything else you would like to discuss, we would be happy to continue the conversation.
> >
> > **References**:
> >
> > [1] Lippe P, Veeling B, Perdikaris P, et al. Pde-refiner: Achieving accurate long rollouts with neural pde solvers[J]. Advances in Neural Information Processing Systems, 2024, 36.
> >
> > [2] Ho J, Jain A, Abbeel P. Denoising diffusion probabilistic models[J]. Advances in neural information processing systems, 2020, 33: 6840-6851.
> >
> > [3] Kohl G, Chen L, Thuerey N. Benchmarking autoregressive conditional diffusion models for turbulent flow simulation[C]. ICML 2024 AI for Science Workshop. 2024.
> >
> > [4] Rühling Cachay S, Zhao B, Joren H, et al. Dyffusion: A dynamics-informed diffusion model for spatiotemporal forecasting[J]. Advances in Neural Information Processing Systems, 2024, 36.
> >
> > [5] Janner M, Du Y, Tenenbaum J, et al. Planning with Diffusion for Flexible Behavior Synthesis[C]. International Conference on Machine Learning. PMLR, 2022: 9902-9915.
> >
> > [6] Yang Z, Su B, Wen J R. Synthesizing long-term human motions with diffusion models via coherent sampling[C]. Proceedings of the 31st ACM International Conference on Multimedia. 2023: 3954-3964.
> >
> > [7] Hwang R, Lee J Y, Shin J Y, et al. Solving pde-constrained control problems using operator learning[C]. Proceedings of the AAAI Conference on Artificial Intelligence. 2022, 36(4): 4504-4512.
> >
> > [8] Wei, Long, Peiyan Hu, Ruiqi Feng, Haodong Feng, Yixuan Du, Tao Zhang, Rui Wang, Yue Wang, Zhi-Ming Ma, and Tailin Wu. A Generative Approach to Control Complex Physical Systems.[C] Advances in Neural Information Processing Systems, 2024, 36.

---

> > > ### Author Response · Authors · 2024-11-25
> > > **A gentle reminder: please respond to our rebuttal**
> > >
> > > Dear Reviewer FYpM,
> > >
> > > Thank you for your time and effort in reviewing our work. We have carefully considered your detailed comments and questions, and we have tried to address all your concerns accordingly.
> > >
> > > As the deadline for author-reviewer discussions is approaching, could you please go over our responses? If you find our responses satisfactory, we hope you could consider adjusting your initial rating. Please feel free to share any additional comments you may have.
> > >
> > > Thank you!
> > >
> > > Authors

---

> > ### Comment · Reviewer_FYpM · 2024-11-25
> >
> > Thanks for the response. I think it clarifies most of the concerns.
> > I think that a score around 7 is appropriate for this work, especially after that the authors made it clear that the model is not able to perform long rollout due to memory constraints, which obviously is a major drawback. The problem is that I have to choose from giving a 6 or 8, so I will wait for the second round comments from other reviewers. I am happy to raise the score to 8 if they deem the work is good.

---

> > > ### Author Response · Authors · 2024-11-25
> > > **Official Response to Reviewer FYpM**
> > >
> > > Thank you for your feedback! We are glad to hear that we have addressed most of your concerns.
> > > - Regarding the issue you raised about the difficulty of performing long rollouts due to memory limitations, we point out that this limitation can be fully addressed with two feasible solutions:
> > >   - Selecting **a noise prediction model with fewer parameters**. Currently, we use U-Net, but the choice of this model is **independent** of our proposed method.
> > >   - Utilizing **multi-GPU** setups to distribute the memory load.
> > > - Regarding the score you mentioned, we gently remind you that in ICLR's scoring system, a score of 6 indicates a "borderline accept", while a score of 8 represents "accept". If you believe our work merits acceptance, we hope you might consider giving us an "accept", as it would greatly support and encourage us!
> > >
> > > If you have any other questions, feel free to reach out to us anytime for further discussion!

---

> > > > ### Comment · Reviewer_FYpM · 2024-11-26
> > > >
> > > > Thanks for the further feedback. I have raised the score to 8 since all reviewers agree that the work is decent.

---

> > > > > ### Author Response · Authors · 2024-11-26
> > > > > **Official Response to Reviewer FYpM**
> > > > >
> > > > > Thank you again for your encouragement and support!

---

> > > > > ### Comment · Reviewer_FYpM · 2024-12-02
> > > > > **Final comment**
> > > > >
> > > > > I have re-read the paper and the discussions between the reviewers and the authors. After considering the grade distribution of ICLR last year, I believe it is still more appropriate to rate this work as a 6 rather than 8.
> > > > >
> > > > > The main issues that restrict me from giving a higher rating have been mentioned in my earlier comments, but I will restate them here for the record:
> > > > >
> > > > > 1. The rollouts of the test cases are actually quite short whilst the authors are adament in claiming that they are long rollouts. It should be mentioned that there are existing works, not only in the field of neural operators but also in other fields, that are able to extend their roll-out horizons to thousands or even tens of thousands of time steps (cf. [1] with graph neural networks, from more than 3 years ago). This is also partially related to the second issue.
> > > > >
> > > > > 2. The authors also mentioned it clearly that the current method is quite memory-heavy, and thus long roll-out with single GPU becomes rather difficult. I understand that neural operators are inherently restricted by the approach itself to be (usually) taking $O(N)$ memory overhead with number of time steps $N$ involved, and I personally do not think that this limitation will nullify the contribution of the work itself, but the fact that the authors are not able to provide a way to circumvent or resolve this limitation is still a major drawback of the work.
> > > > >
> > > > > [1] Pfaff, T. et al. Learning Mesh-Based Simulation with Graph Networks. ICLR 2021

---

> > > > > > ### Author Response · Authors · 2024-12-02
> > > > > > **Official Response to Reviewer FYpM**
> > > > > >
> > > > > > - Thank you for the comment. To address your concerns further, we have provided additional clarifications and experimental results below.
> > > > > > - Regarding the **memory** issue:
> > > > > >   - As mentioned in our response, the memory constraints were addressed **in the context of the reviewer's concern about "the time steps in the 2D experiment being short"**, not suggesting that our method itself is memory-heavy, and certainly not implying that long roll-outs with a single GPU become difficult. As you also noted, increased memory usage in 2D experiments is a **common challenge faced by all neural operator-based methods**.
> > > > > >   - To confirm that our method is not memory-heavy, we tested the **memory usage of U-Net and WDNO in the 2D scenario** you expressed doubts about. The test was conducted on the same GPU with a batch size of 1 during inference. Experiments on 2D Incompressible Fluid and 2D ERA5 datasets show that WDNO has memory usage comparable to U-Net, and the memory requirements are not significant.
> > > > > > |  | 2D Incompressible Fluid | 2D ERA5 |
> > > > > > | --- | --- | --- |
> > > > > > | U-Net | 2234MB | 2402MB |
> > > > > > | WDNO | 2592MB | 2408MB |
> > > > > >   - In our super-resolution experiments, we have demonstrated the prediction of trajectories with **640** timesteps, which was achieved through **a single sampling step on a single GPU**. This further validates that our method is not memory-heavy and can perform long rollouts with a single GPU.
> > > > > >   - Additionally, in our previous response, we **have proposed two feasible approaches** to further reduce memory usage. We point out that **another feasible and straightforward approach to achieving longer-term predictions without increasing memory usage** is to perform multiple rollouts based on the model's own predictions. As mentioned in [1], long-term predictions are also achieved through multiple rollouts, with the longest single-step prediction being 400 steps, which is shorter than our 640-step single prediction.
> > > > > > - Regarding **long-term prediction**:
> > > > > >   - We have already provided the **prediction errors of WDNO and other methods at different timesteps**. The results demonstrate that WDNO has the slowest error growth over time.
> > > > > >   - Additionally, we conducted experiments on the **challenging real-world dataset ERA5** (refer to our responses to Reviewer NMDB, rcJb, and 34r3) to further confirm the advantages of our method in long-term predictions. Instead of the standard 12h-to-12h prediction task, we opted for a more difficult **12h-to-20h** prediction task. In this experiment, WDNO achieved the best results, with a relative $L_2$ error as low as 0.0161.
> > > > > > |  | MSE |
> > > > > > | --- | --- |
> > > > > > | WNO | - |
> > > > > > | MWT | 21.85750 |
> > > > > > | OFormer | 18.26230 |
> > > > > > | FNO | 14.38638 |
> > > > > > | U-Net | 15.51342 |
> > > > > > | DDPM | 15.21103 |
> > > > > > | WDNO | **12.83291** |
> > > > > >
> > > > > > If you have any further concerns, please feel free to reach out. We would be happy to provide additional clarifications.
> > > > > >
> > > > > > [1] Pfaff, T. et al. Learning Mesh-Based Simulation with Graph Networks. ICLR 2021.

---

### Author Response · Authors · 2024-11-21
**General Response**

We thank the reviewers for the thorough reviews and constructive suggestions. We acknowledge the positive comments such as novel (Reviewer rcJb, 34r3), solid (Reviewer rcJb, 34r3, 34r3), technically rigorous (Reviewer rcJb), well-written (Reviewer rcJb, FYpM), and interesting (Reviewer rcJb). We also believe that our proposed WDNO method would significantly contribute to the community.

Based on the reviewers' valuable feedback, we have conducted additional experiments and revised the manuscript, which hopefully resolve the reviewers' concerns. The major additional experiments and improvements are as follows:
- We add experiments on **1D advection equation** and **a challenging real-world dataset ERA5**. From the results, WDNO surpasses all the baselines, which verifies WDNO's strong modeling capability on both simple and complex dynamics. For more details, see responses to Reviewer NMDB, rcJb, and 34r3.
- To provide a more comprehensive comparison and analysis, we evaluate **other strong baselines**, including Transolver, CNO, MSVI, ACDM, and DiffusionPDE, on the 1D compressible Navier-Stokes equation. Additionally, we included **MAE and $L_\infty$ error** results for all methods. WDNO consistently achieves the best performance in both MSE and MAE. For $L_\infty$ error, while the results across methods are closer, WDNO still shows relatively better results. See responses to Reviewer iVpN and rcJb for more details.
- We provide **WNO**'s performance on **zero-shot super resolution**, from which we can observe that WNO does not generalize well to finer resolution. This further confirms the effectiveness of our proposed multi-resolution training. For details, please refer to responses to Reviewer iVpN.
- We **compare WDNO with Fourier transform**, including DDPM in Fourier domain and take FNO as the noise prediction model. The results show that DDPM in the Fourier domain achieves some improvement over the original DDPM but performs far worse than the wavelet transform, highlighting the superiority of wavelet transforms for complex systems with abrupt changes. Additionally, FNO as a noise prediction model performs poorly. For more details, see responses to Reviewer iVpN, NMDB, and FYpM.
- We add ablation studies including evaluating the **influence of measurement noise and number of training samples**. Also, based on previous results in the original submission, we additionally report **errors at different timesteps** and **training times** of all baselines. Results show that WDNO is robust to noise, good at long-term prediction and trains quickly. Please refer to responses to Reviewer rcJb, iVpN and FYpM for details.
- We expand **Related Work** by adding three new paragraphs, making it more comprehensive. More details can be found in responses to Reviewer NMDB.
- To **improve the clarity and structure of the paper**, we add explanatory statements and subheadings, adjust the layout of tables in Appendix, and reorganize Experiments. Also, we redrew **Figure 1** to include more information, provide a high-level overview, and better summarize our method. Additionally, we move the original **Figure 2** to the appendix, add more details, and align it with the training and inference process of the 1D Burgers' equation.

We now address each reviewer's concerns individually. Please see responses below each review.

---

### Comment · Area_Chair_Tkdk · 2024-11-25
**Reviewer responses**

Dear Reviewers,

as the author-reviewer discussion end is approaching, I would strongly encourage you to read the authors' responses and acknowldge so, while also checking if your questions/concerns have been appropriately addressed.

This is a crucial step, as ensures that both reviewers and authors are on the same page, and it also helps us to put in perspective your recommendation.

Thank you again for your time and expertise

Best,

AC

---

### Meta-Review · Area_Chair_Tkdk · 2024-12-17

**Metareview:**

This paper introduces the Wavelet Diffusion Neural Operator (WDNO), a method for simulating and controlling PDE-based physical systems. The authors claim that WDNO addresses limitations of standard diffusion models in handling abrupt changes in trajectory by incorporating wavelet transforms and multi-resolution training. While the proposed approach appears promising, particularly for fluid dynamics problems, the reviewers agree that further benchmarking, with more realistic examples, is necessary to fully assess its capabilities and generalizability.

Also, at least one reviewer raised concerns that the claim regarding long-time statistics was not appropriate, particularly given that recent publications do handle long-term statistics of dynamical systems. Along the same lines, there are some possible issues with the memory footprint for 3D problems with long relaxation times (which require long trajectories to capture the phenomenon of interest). Other reviewers were concerned about how the methodology would generalize to different boundary conditions and geometries, given the difficulty of traditional wavelet methods to handle them. In addition, some of the experimental data seem to be lacking details, and they are not compared against state-of-the-art methods using community metrics (ERA5).

Besides all these issues, the reviewers seem to appreciate the content of the paper, and the authors were responsive and thorough on their responses, so I recommend acceptance.

**Additional Comments On Reviewer Discussion:**

There are several issues on the long-term statistics that were raised after the rebuttal period. When I pointed them, a couple of the reviewers responded and then lowered their scores.

---

### Decision · Program_Chairs · 2025-01-22

Accept (Poster)